# Parameterized Hardness of Zonotope Containment and Neural Network Verification

**Vincent Froese**
Technische Universität Berlin
Algorithmics and Computational Complexity
vincent.froese@tu-berlin.de

**Moritz Grillo**
Max Planck Institute for Mathematics in the Sciences
moritz.grillo@mis.mpg.de

**Christoph Hertrich**
University of Technology Nuremberg
christoph.hertrich@utn.de

**Moritz Stargalla**
University of Technology Nuremberg
moritz.stargalla@utn.de

## Abstract

Neural networks with ReLU activations are a widely used model in machine learning. It is thus important to have a profound understanding of the properties of the functions computed by such networks. Recently, there has been increasing interest in the (parameterized) computational complexity of determining these properties. In this work, we close several gaps and resolve an open problem posted by Froese et al. [COLT '25] regarding the parameterized complexity of various problems related to network verification. In particular, we prove that deciding positivity (and thus surjectivity) of a function $f \colon \mathbb{R}^d \to \mathbb{R}$ computed by a 2-layer ReLU network is W[1]-hard when parameterized by $d$. This result also implies that zonotope (non-)containment is W[1]-hard with respect to $d$, a problem that is of independent interest in computational geometry, control theory, and robotics. Moreover, we show that (a) approximating the maximum within any multiplicative factor in 2-layer ReLU networks, (b) computing the $L_p$-Lipschitz constant for $p \in (0, \infty]$ in 2-layer networks, and (c) approximating the $L_p$-Lipschitz constant in 3-layer networks are all NP-hard and W[1]-hard with respect to $d$. Notably, our hardness results are the strongest known so far and imply that the naive enumeration-based methods for solving these fundamental problems are all essentially optimal under the Exponential Time Hypothesis.

## 1 Introduction

Neural networks with rectified linear unit (ReLU) activations are a common model in deep learning. In practice, such networks are trained on finite datasets and are expected to generalize reliably to unseen inputs. However, even minor perturbations of the input may lead to unexpected or erroneous outputs (Szegedy et al., 2014). This highlights the importance of certification of trained models, which in turn requires a detailed understanding of the functions computed by ReLU networks.

A central problem in this context is *network verification*: Given a subset of inputs $\mathcal{X}$, the question is whether the network's outputs are guaranteed to lie within a prescribed set $\mathcal{Y}$. Commonly, the sets $\mathcal{X}$ and $\mathcal{Y}$ take the form of balls or are specified by linear constraints. This question has received increasing attention in recent years, particularly due to the deployment of neural networks in safety-critical applications (Bojarski et al., 2016; Weng et al., 2018a; Kouvaros & Lomuscio, 2021; Rössig & Petkovic, 2021; Katz et al., 2022). Recently, Froese et al. (2025b) established a connection between the basic verification task to decide whether a 2-layer ReLU network attains a positive output (which is equivalent to surjectivity) and the classical geometry problem of *zonotope containment*. The latter asks whether one zonotope is contained within another, a question that has been extensively studied due to its applications in areas such as robotics and control (Sadraddini & Tedrake, 2019; Gruber & Althoff, 2020; 2021; Kulmburg & Althoff, 2021; Yang et al., 2022; Kulmburg et al., 2025).

Beyond verification, robustness is often a crucial requirement since trained networks are typically expected to be insensitive to small input perturbations. This property is commonly quantified in terms of the network's *Lipschitz constant*, which should ideally be small (Virmaux & Scaman, 2018; Weng et al., 2018b; Fazlyab et al., 2019; Jordan & Dimakis, 2020; Bhowmick et al., 2021; Splittgerber, 2026).

Network verification (Katz et al., 2022; Sälzer & Lange, 2022; Froese et al., 2025b), estimating the Lipschitz constant (Virmaux & Scaman, 2018; Jordan & Dimakis, 2020) and zonotope containment (Kulmburg & Althoff, 2021) are all known to be (co)NP-hard. This intractability is closely linked to the curse of dimensionality: As the input dimension $d$ grows, the search space becomes prohibitively large. A natural follow-up question is whether these problems become tractable for low-dimensional input spaces. This perspective is particularly relevant since, in practice, high-dimensional data is often assumed to lie near a low-dimensional submanifold of the input space. Motivated by this, recent work has studied the *parameterized complexity* of neural network problems such as training (Arora et al., 2018; Froese et al., 2022; Brand et al., 2023; Froese & Hertrich, 2023) and verification (Froese et al., 2025b), see also (Ganian, 2026) for a survey. Notably, while checking injectivity of a 2-layer ReLU network with $n$ hidden neurons can be done in $(d + 1)^d \cdot n^{O(1)}$ time (that is, *fixed-parameter tractability* with respect to $d$) (Froese et al., 2025b), the parameterized complexity status of network verification (in particular positivity) and the Lipschitz constant have been posed as open problems at COLT '25 (Froese et al., 2025a).

## 1.1 OUR CONTRIBUTIONS

We answer the aforementioned questions by proving W[1]-hardness for the parameter input dimension (thus excluding fixed-parameter tractability under standard complexity assumptions). Moreover, we show that solving these problems via simple "brute-force" enumeration of the linear regions of the network's function is essentially optimal under the Exponential Time Hypothesis (ETH).

In Section 3, we give a reduction from the well-known MULTICOLORED CLIQUE problem to network verification in which the network's input dimension depends linearly on the clique size. This reduction forms the basis for our hardness results and yields strong lower bounds based on the ETH. The key difficulty here is that the input dimension must scale linearly with the clique size (in contrast, standard NP-hardness reductions allow the input dimension to grow without restriction).

**Network Verification.** We study the (co)NP-hard problems of deciding positivity, surjectivity, and approximating the maximum of a 2-layer ReLU network $f \colon \mathbb{R}^d \to \mathbb{R}$ (with $n$ hidden neurons), and also the problem of deciding whether a 3-layer ReLU network computes the constant zero function. All these problems are special cases of (complements of) verification. For example, positivity corresponds to checking whether there exists $x \in \mathbb{R}^d$ with $f(x) > 0$, that is, $f(\mathbb{R}^d) \not\subseteq (-\infty, 0]$. All these problems can be solved in $n^{\mathcal{O}(d)} \cdot \mathrm{poly}(N)$ time with simple "brute-force" enumeration algorithms (see Section 2). In Section 4, we prove W[1]-hardness with respect to $d$ for all problems, thereby resolving the open question by Froese et al. (2025a). Our reductions imply a running time lower bound of $n^{\Omega(d)} \cdot \mathrm{poly}(N)$ based on the ETH which shows that the simple enumeration algorithms are essentially optimal. In particular, this implies an $n^{\Omega(d)} \cdot \mathrm{poly}(N)$-time lower bound for the general network verification problem.

**Zonotope Containment.** In Section 5, we study the coNP-hard problem of deciding whether a zonotope $Z \subset \mathbb{R}^d$ (given by its generators) is contained in another zonotope $Z' \subset \mathbb{R}^d$. Based on a duality of 2-layer ReLU networks and zonotopes, we obtain W[1]-hardness with respect to $d$ and an analogous running time lower bound of $n^{\Omega(d)} \cdot \mathrm{poly}(N)$ assuming the ETH which shows that the simple vertex enumeration algorithm is essentially optimal.

**Lipschitz Constant.** Virmaux & Scaman (2018) proved that computing the $L_2$-Lipschitz constant of a 2-layer ReLU network is NP-hard. In Section 6, we extend this to NP-hardness for every $p \in (0, \infty]$ and even W[1]-hardness with respect to $d$. Approximating the $L_p$-Lipschitz constant within any multiplicative constant for 3-layer ReLU networks is known to be NP-hard (Jordan & Dimakis, 2020; Froese et al., 2025b). We also extend this result to W[1]-hardness with respect to $d$. Again, our reductions imply running time lower bounds matching the running times of simple enumeration algorithms. On the positive side, we show that for the restricted class of *input convex* networks,

computing the $L_1$-Lipschitz constant is polynomial-time solvable and the $L_\infty$-Lipschitz constant is *fixed-parameter tractable* (FPT) with respect to $d$. In Section 7, we discuss the equivalence between Lipschitz constant computation and norm maximization on zonotopes and present a randomized FPT-approximation algorithm, using results from subspace embeddings.

**Limitations.** Our paper is clearly of purely theoretical nature. We aim for a thorough understanding of the problems from a computational complexity perspective. Hence, our results are naturally worst-case results. Although the algorithms we give are essentially optimal in terms of running time (assuming the ETH), it might be possible to achieve a better running time by reducing the constant hidden in the exponent. Moreover, additional assumptions on the network structure might render the problems tractable (as in the case of input convex networks for the $L_1$-Lipschitz constant). A full literature review (e.g., for the broad field of network verification) is beyond the scope of this paper.

## 1.2 FURTHER RELATED WORK

Various heuristic methods for network verification have been proposed, including interval bound propagation (Gowal et al., 2018), DeepZ (Wong et al., 2018), DeepPoly (Singh et al., 2019), multi-neuron verification Ferrari et al. (2022), ZonoDual (Jordan et al., 2022), and cutting planes (Zhang et al., 2022). Baader et al. (2024) and Mao et al. (2024) study the expressivity of convex relaxations that are often used in practical network verification algorithms. $L_p$-norm maximization on zonotopes is also known as the *Longest Vector Sum* problem and has a wide range of applications in pattern recognition, clustering, signal processing, and analysis of large-scale data (Baburin & Pyatkin, 2007; Shenmaier, 2018; 2020). Special cases were studied before (Bodlaender et al., 1990; Ferrez et al., 2005).

## 2 PRELIMINARIES

**Notation.** For $n \in \mathbb{N}$, we define $[n] := \{1, \ldots, n\}$. For $k, n \in \mathbb{N}, k \leq n$, we define $\binom{[n]}{k} := \{A \subseteq [n] : |A| = k\}$. A function $f : \mathbb{R}^d \to \mathbb{R}^m$ is *positively homogeneous* if $f(\lambda x) = \lambda f(x)$ holds for all $x \in \mathbb{R}^d$ and $\lambda \geq 0$. Given a *generator* matrix $A = (a_1, \ldots, a_n) \in \mathbb{R}^{d \times n}$, the corresponding *zonotope* is $Z(A) := \sum_{i=1}^{n} \text{conv}(\{0, a_i\})$, where the sum is the Minkowski sum of the generators.

**$L_p$-Lipschitz Constant.** For $p \in (0, \infty)$ and a vector $x \in \mathbb{R}^d$, we define $\|x\|_p := \left( \sum_{i=1}^{d} |x_i|^p \right)^{\frac{1}{p}}$, and for $p = \infty$ we set $\|x\|_\infty := \max_{i \in [d]} |x_i|$. For $p \in [1, \infty]$, the function $\|\cdot\|_p$ is the $L_p$-*norm*, and for $p \in (0, 1)$, it is the $L_p$-*quasinorm*. The $L_0$-function is defined by $\|x\|_0 := |\{i \in [d] : x_i \neq 0\}|$. The $L_p$-Lipschitz constant of a function $f$ is $L_p(f) := \sup_{x \neq y} \frac{\|f(x) - f(y)\|_p}{\|x - y\|_p}$.

**ReLU Networks.** A *ReLU layer* with $d$ inputs, $m$ outputs, weights $W \in \mathbb{R}^{m \times d}$, and biases $b \in \mathbb{R}^m$ computes the map $\phi_{W,b} : \mathbb{R}^d \to \mathbb{R}^m$, $x \mapsto \max(0, Wx + b)$, where the maximum is applied in each component. A *ReLU network* with $\ell \geq 1$ layers and one-dimensional output is defined by $\ell$ weight matrices $W_i \in \mathbb{R}^{n_i \times n_{i-1}}$ and biases $b_i \in \mathbb{R}^{n_i}$ for $i \in [\ell]$, where $n_0 := d, \ldots, n_\ell := 1 \in \mathbb{N}^+$, and computes the *continuous piecewise linear* (CPWL) function $f : \mathbb{R}^d \to \mathbb{R}$ with

$$f(x) := W_\ell \cdot (\phi_{W_{\ell-1}, b_{\ell-1}} \circ \cdots \circ \phi_{W_1, b_1})(x) + b_\ell.$$

Observe that no activation function is applied in the output layer. The $\ell - 1$ ReLU layers are also called *hidden layers*. The *width* and *size* of the network are $\max\{n_1, \ldots, n_{\ell-1}\}$ and $\sum_{i=1}^{\ell-1} n_i$, respectively. Additional details can be found in Appendix A.

**Maxout Networks.** A *2-maxout layer* with $d$ inputs, $m$ outputs, two weight matrices $W^{(1)}, W^{(2)} \in \mathbb{R}^{m \times d}$, and two bias vectors $b^{(1)}, b^{(2)} \in \mathbb{R}^m$ computes the map $\psi_{W^{(1)}, W^{(2)}, b^{(1)}, b^{(2)}} : \mathbb{R}^d \to \mathbb{R}^m$, $x \mapsto \max(W^{(1)}x + b^{(1)}, W^{(2)}x + b^{(2)})$, where the maximum is applied in each component. A *2-maxout network* is defined analogously to a ReLU network, replacing the ReLU layers $\phi_{W_i, b_i}$ with 2-maxout layers $\psi_{W_i^{(1)}, W_i^{(2)}, b_i^{(1)}, b_i^{(2)}}$ for all $i \in [\ell - 1]$. We further note that ReLU networks form a subclass of maxout networks by choosing $W_i^{(1)} = 0$ and

$b_i^{(1)} = 0$ for all $i \in [\ell - 1]$. Moreover, 2-maxout networks can model skip connections using $\max(x, x) = x$.

**Polytopes and Duality.** There is a duality between positively homogeneous convex CPWL functions from $\mathbb{R}^d$ to $\mathbb{R}$ (the set of which is denoted $\mathcal{F}_d$) and polytopes in $\mathbb{R}^d$ (denoted $\mathcal{P}_d$), which we will briefly sketch. Any function $f \in \mathcal{F}_d$ can be written as $f(x) = \max\{a_1^\top x, \ldots, a_k^\top x\}$ for some $a_i \in \mathbb{R}^d$, and its *Newton polytope* is $\mathrm{Newt}(f) \coloneqq \mathrm{conv}(\{a_1, \ldots, a_k\})$. Equivalently, $f$ is the *support function* of $\mathrm{Newt}(f)$, that is, $f(x) = \max_{y \in \mathrm{Newt}(f)} y^\top x$. The map $\varphi \colon \mathcal{F}_d \to \mathcal{P}_d$, defined by $f \mapsto \mathrm{Newt}(f)$, is a bijection satisfying $\varphi(f+g) = \varphi(f) + \varphi(g)$ and $\varphi(\max\{f, g\}) = \mathrm{conv}(\varphi(f) \cup \varphi(g))$, where $+$ denotes pointwise addition or Minkowski sum, respectively.

**Parameterized Complexity.** We assume basic knowledge on computational complexity theory. Parameterized complexity is a multivariate approach to study the time complexity of computational problems (Cygan et al., 2015; Downey & Fellows, 2013). A *parameterized problem* $L \subseteq \Sigma^* \times \mathbb{N}$ consists of instances $(x, k)$ where $x$ encodes a classical problem instance and $k$ is a *parameter*. A parameterized problem $L$ is *fixed-parameter tractable* (contained in the class FPT) if it can be solved in $f(k) \cdot |x|^{\mathcal{O}(1)}$ time, where $f$ is an arbitrary function that only depends on $k$. The class XP contains all parameterized problems which can be solved in polynomial time for constant parameter values, that is, in $f(k) \cdot |x|^{g(k)}$ time, where $g$ is an arbitrary function that only depends on $k$. It is known that FPT $\subsetneq$ XP. The class W[1] can be defined as the set of all parameterized problems which can be reduced to CLIQUE (with parameter solution size) via a *parameterized reduction*. It is known that FPT $\subseteq$ W[1] $\subseteq$ XP and it is widely believed that W[1] contains problems which are not in FPT (namely the W[1]-hard problems such as CLIQUE). A parameterized reduction from $L$ to $L'$ is an algorithm that maps an instance $(x, k)$ in $f(k) \cdot |x|^{\mathcal{O}(1)}$ time to an instance $(x', k')$ such that $k' \le g(k)$ for an arbitrary function $g$ and $(x, k) \in L$ if and only if $(x', k') \in L'$.

The *Exponential Time Hypothesis* (Impagliazzo & Paturi, 2001) states that 3-SAT on $n$ variables cannot be solved in $2^{o(n)}$ time. The ETH implies FPT $\ne$ W[1] (which implies P $\ne$ NP), as well as running time lower bounds: For example, CLIQUE cannot be solved in $\rho(k) \cdot n^{o(k)}$ time, where $k$ is the size of the requested clique and $n$ is the number of nodes in the graph (Cygan et al., 2015).

## 2.1 PROBLEM DEFINITIONS AND WARM-UP

For given generator matrices $A \in \mathbb{R}^{d \times n}$ and $B \in \mathbb{R}^{d \times m}$, and a scalar $L \in \mathbb{R}$, we consider the following problems:

- ZONOTOPE CONTAINMENT: Is $Z(A) \subseteq Z(B)$?
- $L_p$-MAX ON ZONOTOPES: Is $\max_{x \in Z(A)} \|x\|_p \ge L$?

For an $\ell$-layer ReLU network defined by weight matrices $W_i \in \mathbb{R}^{n_i \times n_{i-1}}$ and biases $b_i \in \mathbb{R}^{n_i}$ for $i \in [\ell]$, where $d \coloneqq n_0, \ldots, n_\ell \coloneqq 1 \in \mathbb{N}^+$ that computes the function $f \colon \mathbb{R}^d \to \mathbb{R}$, $f(x) \coloneqq W_\ell \cdot (\phi_{W_{\ell-1}, b_{\ell-1}} \circ \cdots \circ \phi_{W_1, b_1})(x) + b_\ell$, we consider the following problems:

- $\ell$-LAYER RELU POSITIVITY: Is there an $x \in \mathbb{R}^d$ such that $f(x) > 0$?
- $\ell$-LAYER RELU SURJECTIVITY: Is $f$ surjective (that is, $\forall y \in \mathbb{R} \, \exists x \in \mathbb{R}^d : f(x) = y$)?
- $\ell$-LAYER RELU $L_p$-LIPSCHITZ CONSTANT: Is $L_p(f) \ge L$?

In fact, all these problems are known to be in XP for the parameter $d$ (simply enumerate vertices of zonotopes and linear regions of ReLU networks; see Appendix B for more details).

**Theorem 2.1.** ZONOTOPE CONTAINMENT *and* $L_p$-MAX ON ZONOTOPES *can be solved in* $\mathcal{O}(n^{d-1} \cdot \mathrm{poly}(N))$ *time (where $n$ is the number of generators and $N$ is the input bit-length).*

**Theorem 2.2.** $\ell$-LAYER RELU POSITIVITY, $\ell$-LAYER RELU SURJECTIVITY, $\ell$-LAYER RELU $L_p$-LIPSCHITZ CONSTANT, *computing the maximum of an $\ell$-layer ReLU network over a polyhedron and deciding whether an $\ell$-layer ReLU network computes the zero function can be solved in* $\mathcal{O}(n^{(\ell-1)d} \cdot \mathrm{poly}(N))$ *time (where $n$ is the network width and $N$ is the input bit-length).*

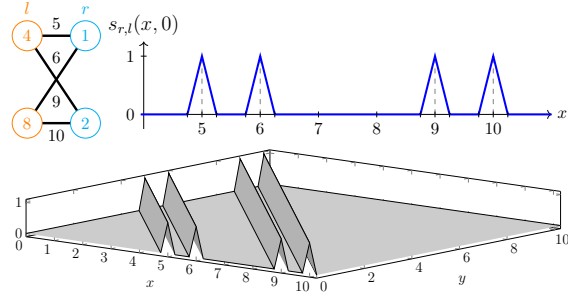
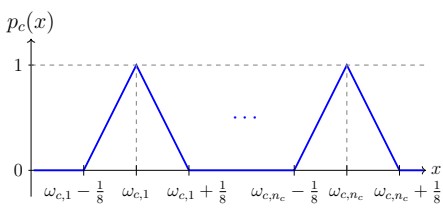

Figure 1: Spike function $s_{r,l}$ for a colored graph (top left). Node labels: $\omega_{r,1} = 1, \omega_{r,2} = 2, \omega_{l,1} = 4, \omega_{l,2} = 8$. Edge labels: $\omega_{r,1,l,1} = 5, \omega_{r,2,l,1} = 6, \omega_{r,1,l,2} = 9, \omega_{r,2,l,2} = 10$.

Figure 2: Penalty function $p_c$.

In particular, we prove in Appendix B that network verification for $\ell$-layer ReLU networks $f : \mathbb{R}^d \to \mathbb{R}^m$ is solvable in $\mathcal{O}(n^{(\ell-1)d} \cdot \text{poly}(N))$ time, assuming that $\mathcal{X}$ and $\mathcal{Y}$ are polyhedra in halfspace representation. Later, we will prove that, assuming the ETH, the 2-layer or the 3-layer versions of all of these problems cannot be solved in $\rho(d) \cdot N^{o(d)}$ time for any function $\rho$, which means that the $\mathcal{O}(n^d \cdot \text{poly}(N))$- and $\mathcal{O}(n^{2d} \cdot \text{poly}(N))$-time algorithms (for 2- and 3-layer networks) are essentially optimal with respect to the runtime dependency on $d$. Note that hardness results for 2- or 3-layer networks also imply hardness for deeper networks with $\ell \geq 3$ layers: simply concatenate the 2- or 3-layer network with trivial additional layers that compute the identity map.

## 3 REDUCTION FROM MULTICOLORED CLIQUE

In this section, we present a parameterized reduction which forms the basis for the hardness results for all our considered problems. (All proofs that are omitted from the main text as well as some auxiliary statements can be found in Appendix B.) We reduce from the following problem.

MULTICOLORED CLIQUE

**Input:** A graph $G = (V = V_1 \dot\cup \cdots \dot\cup V_k, E)$, where each node in $V_i$ has color $i$.
**Question:** Does $G$ have a $k$-colored clique (a clique with exactly one node of each color)?

MULTICOLORED CLIQUE is NP-hard, W[1]-hard with respect to $k$ and not solvable in $\rho(k) \cdot |V|^{o(k)}$ time for any computable function $\rho$ assuming the ETH (Cygan et al., 2015).

**Proposition 3.1.** *For every instance $(G = (V = V_1 \dot\cup \cdots \dot\cup V_k, E), k)$ of* MULTICOLORED CLIQUE, *it is possible to construct in polynomial time a 2-layer ReLU network computing a function $f : \mathbb{R}^k \to \mathbb{R}$ such that $\max_{x \in \mathbb{R}^k} f(x) = k + \binom{k}{2}$ if and only if $G$ contains a $k$-colored clique and $\max_{x \in \mathbb{R}^k} f(x) \leq k + \binom{k}{2} - 1$ otherwise.*

*Proof Sketch.* Let $(G = (V = V_1 \dot\cup \cdots \dot\cup V_k, E), k)$ be an instance of MULTICOLORED CLIQUE, where $V_c = \{v_{c,1}, \ldots, v_{c,n_c}\}$ and $E = \bigcup_{(r,l) \in \binom{[k]}{2}} E_{r,l}$, where $E_{r,l}$ denotes the set of edges whose nodes have color $r$ and $l$. We assign each node $v_{c,i}$ a unique label $\omega_{c,i} \in \mathbb{N}$ such that every edge $\{v_{r,i}, v_{l,j}\}$ gets a unique label $\omega_{r,i,l,j} := \omega_{r,i} + \omega_{l,i}$ (using Sidon sets, see Appendix B for details).

For every color pair $(r, l) \in \binom{[k]}{2}$, we introduce a *spike function* $s_{r,l} : \mathbb{R}^2 \to [0, 1]$ (see Figure 1) that is zero everywhere except for a set of $|E_{r,l}|$ parallel stripes in which $s_{r,l}$ forms a spike, that is, goes up from 0 to 1 and goes down from 1 to 0 again. The spike function attains value 1 if and only if the sum of its inputs is equal to $\omega_{r,i,l,j}$ for some edge $\{v_{r,i}, v_{l,j}\} \in E_{r,l}$. The spike function can be implemented with $3|E_{r,l}|$ neurons. For every color $c \in [k]$, we create a *penalty function* $p_c : \mathbb{R} \to [0, 1]$ (see Figure 2) that has a narrow spike around the value $\omega_{c,i}$ for each node $v_{c,i}$ and is zero everywhere else. The penalty function $p_c$ can be implemented with $3n_c$ neurons.

By computing all spike and penalty functions in parallel and summing them up, we obtain a 2-layer ReLU network with $3(|V| + |E|)$ ReLU neurons that computes $f : \mathbb{R}^k \to [0, k + \binom{k}{2}]$ with

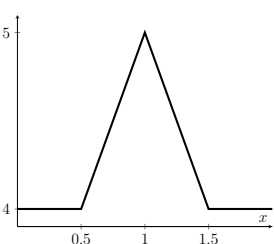 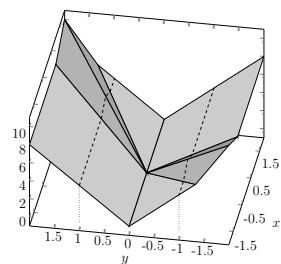 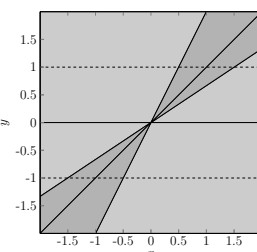

Figure 3: Homogenization: the function $\max(0, 2x - 1) - \max(0, 4x - 4) + \max(0, 2x - 3) + 4$ (left) is turned into $\max(0, 2x - y) - \max(0, 4x - 4y) + \max(0, 2x - 3y) + 4|y|$ (right).

$$f(x_1, \ldots, x_k) = \sum_{(r,l) \in \binom{[k]}{2}} s_{r,l}(x_r, x_l) + \sum_{c \in [k]} p_c(x_c).$$

Next, we show that if there exists a $k$-colored clique $\{v_{1,a_1}, \ldots, v_{k,a_k}\}$ in $G$, then $f((\omega_{1,a_1}, \ldots, \omega_{k,a_k})) = k + \binom{k}{2}$. On the other hand, we show that if there is a point $x^* \in \mathbb{R}^k$ with $f(x^*) > k + \binom{k}{2} - 1$, then $G$ has a $k$-colored clique. The idea is that in this case, all spike and penalty functions must have positive output. For the penalty functions, this means that every input value $x_c^*$ must be close to a value $\omega_{c,a_c}$ which corresponds to the node $v_{c,a_c}$. Since the spike functions only give a positive output if the two node inputs correspond to adjacent nodes, the nodes $v_{1,a_1}, \ldots, v_{k,a_k}$ then form a $k$-colored clique in $G$. $\qquad\square$

In the following, we will use modifications of this construction to prove our hardness results. All our (parameterized) reductions are in fact *polynomial-time reductions* and thus also prove NP-hardness. We will only state this explicitly if the NP-hardness of the problem was not previously known.

## 4 Hardness of Network Verification Problems

We first prove W[1]-hardness (w.r.t. $d$) of 2-LAYER RELU POSITIVITY. The NP-hardness of 2-LAYER RELU POSITIVITY was established by Froese et al. (2025b). We prove W[1]-hardness via the reduction from Proposition 3.1, which relies on the use of biases. To extend the hardness result to other problems, we need to show a stronger statement: that 2-LAYER RELU POSITIVITY remains W[1]-hard even when all biases are equal to zero. For this, we use *homogenized* ReLU networks.

**Definition 4.1.** Given a 2-layer ReLU network with a single output neuron, its *homogenization* is the ReLU network (with all biases equal to zero) that is obtained by adding an extra input variable $y$ to the network, replacing all biases $b$ of neurons in the first hidden layer by $y \cdot b$ and replacing the bias $b$ of the output neuron by $|y| \cdot b$ using two extra neurons in the hidden layer.

Figure 3 illustrates the effect of homogenization on the function of a 2-layer ReLU network.

**Theorem 4.2.** 2-LAYER RELU POSITIVITY *is W[1]-hard with respect to $d$ and not solvable in $\rho(d) \cdot N^{o(d)}$ time (where $N$ is the input bit-length) for any function $\rho$ assuming the ETH, even if all biases are zero.*

*Proof Sketch.* Setting the output node bias of the ReLU network constructed in the proof of Proposition 3.1 to $1 - k - \binom{k}{2}$ yields a network that has a positive output if and only if the graph $G$ from the MULTICOLORED CLIQUE instance contains a $k$-colored clique. We then show that homogenizing this network preserves this equivalence, which yields a parameterized reduction from MULTICOLORED CLIQUE to 2-LAYER RELU POSITIVITY without biases (and thus proves W[1]-hardness). Note that the input dimension $d$ of the constructed network is $k + 1$. Hence, any algorithm solving 2-LAYER RELU POSITIVITY in $\rho(d) \cdot N^{o(d)}$ time would imply an algorithm for MULTICOLORED CLIQUE running in $\rho(k) \cdot |V|^{o(k)}$ time (since $N \leq |V|^{\mathcal{O}(1)}$) contradicting the ETH. $\qquad\square$

Theorem 4.2 also implies W[1]-hardness w.r.t. the input dimension $d$ for approximating the maximum of a 2-layer ReLU network over a polyhedron within any multiplicative factor. Froese et al. (2025b, Corollary 13) showed that approximating this value is NP-hard.

**Corollary 4.3.** *Approximating the maximum of a 2-layer ReLU network over a polyhedron within any multiplicative factor is W[1]-hard with respect to its input dimension $d$ and cannot be done in $\rho(d) \cdot N^{o(d)}$ time (where $N$ is the input bit-length) for any function $\rho$ assuming the ETH.*

By adding another hidden layer with a single ReLU neuron to the network constructed in the proof of Theorem 4.2, we obtain a 3-layer ReLU network that has a non-zero output if and only if the original 2-layer network has a positive output. This yields the following corollary.

**Corollary 4.4.** *The problem of deciding whether a 3-layer ReLU network computes a non-zero function is W[1]-hard with respect to its input dimension $d$ and not solvable in $\rho(d) \cdot N^{o(d)}$ time (where $N$ is the input bit-length) for any function $\rho$ assuming the ETH.*

The NP-hardness of the above problem was established by Froese et al. (2025b). For 2-layer networks, it is solvable in polynomial time (Froese et al., 2025b), which holds also in the presence of biases (Stargalla et al., 2025). Thus, Corollary 4.4 draws an even clearer boundary between the computational complexity of this problem in the 2-layer and 3-layer cases.

Froese et al. (2025b) proved NP-hardness of 2-LAYER RELU SURJECTIVITY and asked whether the problem is fixed-parameter tractable with respect to $d$. We give a negative answer to this question.

**Theorem 4.5.** 2-LAYER RELU SURJECTIVITY *is $W[1]$-hard with respect to $d$ and not solvable in $\rho(d) \cdot N^{o(d)}$ time (where $N$ is the input bit-length) for any function $\rho$ assuming the ETH.*

*Proof.* Recall that a positively homogeneous function $g \colon \mathbb{R}^d \to \mathbb{R}$ is surjective if and only if there exist two points $v^+, v^- \in \mathbb{R}^d$ such that $g(v^+) > 0$ and $g(v^-) < 0$. The positively homogeneous function $f \colon \mathbb{R}^{k+1} \to \mathbb{R}$ of the 2-layer ReLU network from the proof of Theorem 4.2 is in fact surjective if and only if it has a positive point, as $f(\mathbf{0}, 1) < 0$. $\qquad \square$

## 5 HARDNESS OF ZONOTOPE NON-CONTAINMENT

In this section, we prove W[1]-hardness for ZONOTOPE NON-CONTAINMENT, the complement of ZONOTOPE CONTAINMENT. ZONOTOPE CONTAINMENT is coNP-complete, and can be solved in $O(n^{d-1} \cdot \mathrm{poly}(N))$ time (where $N$ is the input bit-length) by enumerating the vertices of one zonotope, but fixed-parameter tractability with respect to the dimension $d$ remained open so far (Froese et al., 2025a). Moreover, Kulmburg & Althoff (2021) showed that containment is equivalent to maximizing a certain zonotope norm, making it a special case of norm maximization on zonotopes.

Froese et al. (2025b) showed that 2-LAYER RELU POSITIVITY is equivalent to ZONOTOPE NON-CONTAINMENT following from the duality of positively homogeneous convex CPWL functions from $\mathbb{R}^d$ to $\mathbb{R}$ and polytopes in $\mathbb{R}^d$. We will briefly sketch this equivalence. Let a 2-layer ReLU network without biases be given by $g(x) = \sum_{i=1}^m \lambda_i \max\{0, w_i^\top x\}$. We can assume without loss of generality that $\lambda_i \in \{-1, 1\}$ due to the positive homogeneity of $\max\{0, w_i^\top x\}$. Hence, $g(x) = \sum_{i \in I^+} \max\{0, w_i^\top x\} - \sum_{i \in I^-} \max\{0, w_i^\top x\}$ for vectors $w_i \in \mathbb{R}^d, i \in I^+ \cup I^-$. Defining the zonotopes

$$Z^+ := \varphi(\sum_{i \in I^+} \max\{0, w_i^\top x\}) = \sum_{i \in I^+} \mathrm{conv}(\{0, w_i\}),$$

$$Z^- := \varphi(\sum_{i \in I^-} \max\{0, w_i^\top x\}) = \sum_{i \in I^-} \mathrm{conv}(\{0, w_i\}),$$

it holds that $g = \varphi^{-1}(Z^+) - \varphi^{-1}(Z^-)$. By definition of the support function, $Z^+ \subseteq Z^-$ implies $\varphi^{-1}(Z^+) \leq \varphi^{-1}(Z^-)$. Conversely, if $y_* \in Z^+ \setminus Z^-$, then there is a separating hyperplane $H = \{y \in \mathbb{R}^d : y^\top x = b\}$ such that $y_*^\top x > b$ and $y^\top x < b$ for all $y \in Z^-$. Hence, $g(x) > 0$ (see Figure 4 for an illustration). Since also any pair of zonotopes is of this form, 2-LAYER RELU POSITIVITY is equivalent to ZONOTOPE NON-CONTAINMENT. Thus, Theorem 4.2 implies the following theorem.

**Theorem 5.1.** ZONOTOPE NON-CONTAINMENT *is $W[1]$-hard with respect to $d$ and not solvable in $\rho(d) \cdot N^{o(d)}$ time (where $N$ is the input bit-length) for any function $\rho$ assuming the ETH.*

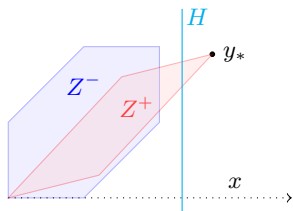

Figure 4: An illustration of the equivalence between 2-LAYER RELU POSITIVITY and ZONOTOPE NON-CONTAINMENT. Let $H = \{y \in \mathbb{R}^d \mid y^\top x = b\}$ be a hyperplane that separates $y_*$ from $Z^-$. Then $g(x) = \max_{y \in Z^+} y^\top x - \max_{y \in Z^-} y^\top x > y_*^\top x - b > 0$.

## 6  HARDNESS OF COMPUTING THE LIPSCHITZ CONSTANT

Jordan & Dimakis (2020) established the NP-hardness for approximating the $L_p$-Lipschitz constant for $p = 1$ and $p = \infty$ for 3-layer ReLU networks within a multiplicative factor of $\Omega(N^{1-\varepsilon})$ for every constant $\varepsilon > 0$, where $N$ is the encoding size of the ReLU network. The NP-hardness result of Froese et al. (2025b) for 2-LAYER RELU POSITIVITY implies NP-hardness for approximating the $L_p$-Lipschitz constant for $p \in [0, \infty]$ within any multiplicative factor for 3-layer ReLU networks. We extend this by showing W[1]-hardness of the problem.

**Corollary 6.1.** *For all $p \in [0, \infty]$, approximating the $L_p$-Lipschitz constant of a 3-layer ReLU network by any multiplicative factor is W[1]-hard with respect to its input dimension $d$ and cannot be done in $\rho(d) \cdot N^{o(d)}$ time (where $N$ is the input bit-length) for any function $\rho$ assuming the ETH.*

*Proof.* Adding a hidden layer with a single ReLU neuron to the construction in the proof of Theorem 4.2 yields a 3-layer network which computes a function with a non-zero $L_p$-Lipschitz constant if and only if the original 2-layer network has a positive output. Hence, any multiplicative approximation could be used to decide 2-LAYER RELU POSITIVITY. □

Virmaux & Scaman (2018) established the NP-hardness of 2-LAYER RELU $L_2$-LIPSCHITZ CONSTANT. We extend the NP-hardness to $p \in (0, \infty]$ and show W[1]-hardness w.r.t. $d$.

**Theorem 6.2.** *For all $p \in (0, \infty]$, 2-LAYER RELU $L_p$-LIPSCHITZ CONSTANT is NP-hard, W[1]-hard with respect to $d$ and not solvable in $\rho(d) \cdot N^{o(d)}$ time (where $N$ is the input bit-length) for any function $\rho$ assuming the ETH.*

*Proof Sketch.* First, we show that for any positively homogeneous CPWL function $f \colon \mathbb{R}^d \to \mathbb{R}$, we have $L_p(f) = \max_{\|x\|_p \leq 1} |f(x)|$. The idea is now to scale all $y$ coefficients of the function $g \colon \mathbb{R}^{k+1} \to \mathbb{R}$ computed by the homogenized network constructed in the proof of Proposition 3.1 by a sufficiently small amount $\varepsilon$ to obtain the positively homogeneous CPWL function $h \colon \mathbb{R}^{k+1} \to \mathbb{R}$. Then, every $x^* \in \arg\max_{x \in \mathbb{R}^k} h(x, 1)$ has (sufficiently) small entries, as scaling the $y$ coefficients is equivalent to scaling the spike and penalty functions. We then show that $L_p(h)$ is almost equal to $\mathcal{L} := \max_{x \in \mathbb{R}^k} h(x, 1)$, as we can scale down a maximizer $x^* \in \arg\max_{x \in \mathbb{R}^k} h(x, 1)$ with a $y^*$ that is only slightly smaller than 1 to obtain a feasible point $y^*(x, 1)$ for $\max_{\|(x,y)\|_p \leq 1} |h(x, y)|$ with value $|h(y^* \cdot x, y^*)| = y^*|h(x^*, 1)|$, which proves $\mathcal{L} \geq L_p(h) \geq \mathcal{L} \cdot y^*$ (so $L_p(h) \approx \mathcal{L}$). We conclude the proof by showing that the hardness of computing $\mathcal{L}$ transfers to computing $L_p(h)$. □

On the positive side, we show that for a special subclass of ReLU networks, computing the $L_1$- and $L_\infty$-Lipschitz constant is tractable.

**Input Convex Neural Networks.**  A 2-maxout network is *input-convex* (ICNN) if the weight matrices of all but the first layer have only nonnegative entries, resulting in a convex function $f(x) = \max\{a_1^\top x + b_1, \ldots, a_k^\top x + b_k\}$. The $L_p$-Lipschitz constant of $f$ is given by the maximum, taken over all linear regions $C$ of $f$, of the $L_p$-Lipschitz constant of $f$ restricted to $C$, where $f(x) = a_C^\top x + b_C$ for all $x \in C$. Using the well-known equality $L_p(g) = \max_{x \in \mathbb{R}^d} \|\nabla g(x)\|_q$ for smooth functions $g \colon \mathbb{R}^d \to \mathbb{R}$ (Jordan & Dimakis, 2020), we derive that the $L_p$-Lipschitz constant of $f$ restricted to

the region $C$ is equal to $\|a_C\|_q$ and thus $L_p(f) = \max_{C \text{ linear region of } f} \|a_C\|_q = \max_i \|a_i\|_q$, where $\|\cdot\|_q$ is the *dual* norm of the $L_p$-norm. Note that the function $f$ of the ICNN has the same $L_p$-Lipschitz constant as the function $g(x) = \max\{a_1^\top x, \ldots, a_k^\top x\}$ computed by the same network where all biases are set to 0, which implies that we might assume without loss of generality that the network does not have biases and hence computes a function $f$ that is convex and positively homogeneous.

Hertrich & Loho (2024) showed that there is a small *extended formulation* of $\mathrm{Newt}(f)$ for a function $f$ computed by an ICNN without biases. More precisely, their proofs reveal that for a function $f: \mathbb{R}^d \to \mathbb{R}$ computed by an ICNN, there is a polytope $Q \subseteq \mathbb{R}^{d+m}$ and a projection $\pi: \mathbb{R}^{d+m} \to \mathbb{R}^d$ such that $\pi(Q) = \mathrm{Newt}(f)$ and the encoding size of $(Q, \pi)$ is polynomial in the encoding size of $f$, where $Q$ is given in half-space representation. Using this, we prove the following proposition.

**Proposition 6.3.** *Let $f: \mathbb{R}^d \to \mathbb{R}$ be an ICNN with encoding size $N$. Then $L_1(f)$ can be computed in $\mathrm{poly}(N)$ time and $L_\infty(f)$ can be computed in $O(2^d \mathrm{poly}(N))$ time.*

*Proof.* By the discussion above, we can assume without loss of generality that there are no biases and $f$ is positively homogeneous. In this case, the definition of the support function implies that $L_p(f) = \max_{y \in \mathrm{Newt}(f)} \|y\|_q$. By Hertrich & Loho (2024), there exists a polytope $Q$ and a projection $\pi$ with $\mathrm{poly}(N)$ encoding size such that $L_p(f) = \max_{y \in Q} \|\pi(y)\|_q$. For $p = \infty$ and $p = 1$, this maximization can be reduced to finitely many LPs: Indeed, $\max_{y \in Q} \|\pi(y)\|_\infty = \max_{c \in \{\pm e_1, \ldots, \pm e_d\}} \max_{y \in Q} c^\top \pi(y)$, which requires solving only $2d$ LPs, while $\max_{y \in Q} \|\pi(y)\|_1 = \max_{c \in \{\pm 1\}^d} \max_{y \in Q} c^\top \pi(y)$, which requires solving $2^d$ LPs. Since LPs can be solved in polynomial time, the statements follow. $\square$

## 7 NORM MAXIMIZATION ON ZONOTOPES

We close with a short section describing a connection between Lipschitz constants of neural networks and norm maximization on zonotopes. For 2-layer ICNNs $f: \mathbb{R}^d \to \mathbb{R}$, we can restrict ourselves without loss of generality to the case where all output weights are equal to 1. In this case, the Newton polytope $\mathrm{Newt}(f)$ is a zonotope and computing $L_p(f)$ is equivalent to maximizing the dual norm of the $L_p$-norm over this zonotope (see Froese et al. (2025a)).

Baburin & Pyatkin (2007) showed that maximizing the $L_\infty$-norm on zonotopes is solvable in polynomial time and maximizing the $L_1$-norm on zonotopes is fixed-parameter tractable for $d$ (our Proposition 6.3 generalizes these results). Note that Theorem 5.1 implies that maximizing a zonotope-norm over a zonotope is W[1]-hard with respect to the dimension $d$ (since zonotope containment is equivalent to this problem (Kulmburg & Althoff, 2021)). For $p \in (1, \infty)$, however, it is an open question whether $L_p$-maximization on zonotopes is fixed-parameter tractable for $d$ (Froese et al., 2025a). Shenmaier (2018) proved NP-hardness and inapproximability for $p \in [1, \infty)$ and showed a randomized (sampling based) $(1 - \varepsilon)$-approximation with probability $1 - 1/\varepsilon$ in time $(1 + 2/\varepsilon)^d \mathrm{poly}(d, n)$ for every $\varepsilon \in (0, 1)$ and an arbitrary norm. We show that known results from *subspace embedding* theory can also be used to obtain randomized approximations, which is an interesting application of these results. The worst-case running time, however, is worse, but in practice the actual running time might still be faster. Bozzai et al. (2023) observed that results for $\ell_1$ subspace embeddings (Cohen & Peng, 2015) yield *zonotope order reductions*, that is, approximations of zonotopes with few generators. More precisely, the following can be derived.

**Theorem 7.1.** *There is a polynomial-time algorithm which, given a matrix $A \in \mathbb{R}^{d \times n}$ and $\varepsilon > 0$, outputs a matrix $A' \in \mathbb{R}^{d \times r}$ with $r \in \mathcal{O}(d \log d \varepsilon^{-2})$ such that with high probability*

$$(1 + \varepsilon)^{-1} Z(A') \subseteq Z(A) \subseteq (1 + \varepsilon) Z(A').$$

This order reduction yields a simple randomized approximation algorithm.

**Theorem 7.2.** *Let $\|\cdot\|$ be any norm on $\mathbb{R}^d$ (computable in time $T$). There is a randomized algorithm which, given a matrix $A \in \mathbb{R}^{d \times n}$ and $\varepsilon > 0$, outputs a value $\alpha \in \mathbb{R}$ in $\mathcal{O}((cd \log d / \varepsilon^2)^{d-1} \cdot T + \mathrm{poly}(n))$ time (for some constant $c > 0$) such that with high probability*

$$(1 + \varepsilon)^{-1} \alpha \leq \max_{x \in Z(A)} \|x\| \leq (1 + \varepsilon) \alpha.$$

*Proof.* Note that every norm is convex and convex functions attain their maximum on a polytope at a vertex. On input $(A, \varepsilon)$, we run the algorithm from Theorem 7.1 to obtain a matrix $A'$ with $r \in \mathcal{O}(d \log d\varepsilon^{-2})$ columns in polynomial time. The zonotope $Z(A')$ has at most $\mathcal{O}(r^{d-1})$ vertices (Zaslavsky, 1975), which can be enumerated in $\mathcal{O}(r^{d-1})$ time (Ferrez et al., 2005). We simply return the maximum $\|\cdot\|$-value $\alpha$ of these vertices. Then, with high probability, it holds $(1 + \varepsilon)^{-1} Z(A') \subseteq Z(A) \subseteq (1 + \varepsilon) Z(A')$, which implies

$$\max_{x \in (1+\varepsilon)^{-1} Z(A')} \|x\| = (1+\varepsilon)^{-1}\alpha \leq \max_{x \in Z(A)} \|x\| \leq (1+\varepsilon)\alpha = \max_{x \in (1+\varepsilon)Z(A')} \|x\|,$$

due to absolute homogeneity of norms. $\square$

## 8 CONCLUSION

We proved the strongest hardness results for various computational problems related to ReLU network verification known so far. Note that nearly all considered problems can be phrased in terms of maximizing a certain norm over a zonotope; a problem with numerous applications in other areas. Most importantly, our results imply that simple "brute-force" enumeration algorithms are basically best possible with respect to the dependency of the running time on the input dimension. Thus, we settled the parameterized complexity of a wide range of problems almost completely. Moreover, our results show that it does not help to assume that the network weights are sparse and small, since our constructions use only a constant number of (polynomially bounded) non-zero weights for each ReLU neuron. It is thus not easy to formulate a general guidance to circumvent this hardness in practice. One would have to make very specific assumptions on the network structure to ensure that the number of linear regions is small and easy to enumerate. It is not clear which assumptions would be natural here and whether networks trained on real-world data satisfy them. Alternatively, one might use techniques (possibly incorporated into the training process) that guarantee efficient verification or use special architectures (such as ICNNs). We also discussed some tractable cases for restricted subclasses of problems as well as a randomized FPT-approximation. Overall, our hardness results prove and justify that such techniques and the use of heuristics are indeed required in practice to achieve reasonable running times.

The most prominent open question is the fixed-parameter tractability of $L_p$-maximization on zonotopes for $p \in (1, \infty)$ when parameterized by $d$. Recall that this is equivalent to 2-LAYER RELU $L_p$-LIPSCHITZ CONSTANT with only positive output weights. As a first step, one might try to find a deterministic FPT-approximation for norm maximization on zonotopes (e.g., by derandomizing the subspace embedding approach). Also, the complexity of computing $L_0(f)$ for 2-layer ReLU networks is open. Another interesting question is whether running times in $\mathcal{O}(n^{cd})$ for $c < 1$ can be achieved for the considered problems. Last but not least, we wonder whether 2-LAYER RELU POSITIVITY is contained in W[1]. This would settle the parameterized complexity completely.

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
