# Supplemental:
# Parameterized Hardness of Zonotope Containment and Neural Network Verification

## A   Additional Material

### A.1   Additional Preliminaries

**Geometry of ReLU Networks.**   We repeat basic definitions from polyhedral geometry, see Schrijver (1986) for more details. A *polyhedron $P$* is the intersection of finitely many closed halfspaces. A *polyhedral cone $C \subseteq \mathbb{R}^d$* is a polyhedron such that $\lambda u + \mu v \in C$ for every $u, v \in C$ and $\lambda, \mu \in \mathbb{R}_{\geq 0}$. A cone is *pointed* if it does not contain a straight line. A *ray $\rho$* is a one-dimensional pointed cone; a vector $r$ is a *ray generator* of $\rho$ if $\rho = \{\lambda r : \lambda \geq 0\}$. A *polyhedral complex $\mathcal{P}$* is a finite collection of polyhedra such that $\emptyset \in \mathcal{P}$, if $P \in \mathcal{P}$, then all faces of $P$ are in $\mathcal{P}$, and if $P, P' \in \mathcal{P}$, then $P \cap P'$ is a face of $P$ and $P'$. A *cell* is a full-dimensional element of a polyhedral complex. A *hyperplane arrangement $\mathcal{H}$* is a finite collection of hyperplanes in $\mathbb{R}^d$.

A *ReLU network* computing the CWPL function $f \colon \mathbb{R}^d \to \mathbb{R}$ with

$$f(x) := W_\ell \cdot (\phi_{W_{\ell-1}, b_{\ell-1}} \circ \cdots \circ \phi_{W_1, b_1})(x) + b_\ell,$$

is affine linear on each cell of an associated polyhedral complex $\Sigma_f$ (the one induced by the closure of the full-dimensional activation regions (Hanin & Rolnick, 2019) of the network). In particular, the encoding size of every polyhedron in this complex is polynomially bounded in the encoding size of the neural network (Froese et al., 2025b). It is well known that for 2-layer ReLU networks, $\Sigma_f$ corresponds to a hyperplane arrangement (Montúfar et al., 2014).

**Hyperplane Arrangements.**   Any hyperplane arrangement with $n$ hyperplanes in $d$ dimensions has at most $\mathcal{O}(n^d)$ many cells (Zaslavsky, 1975), and it is possible to enumerate them in $\mathcal{O}(n^d)$ time (Edelsbrunner et al., 1986). All zero- and one-dimensional faces of a hyperplane arrangement can be enumerated in $\mathcal{O}(n^d \cdot \mathrm{poly}(N))$ and $\mathcal{O}(n^{d-1} \cdot \mathrm{poly}(N))$ time, respectively, where $N$ is the encoding size of the hyperplane arrangement, since they arise from an intersection of $d$ and $d-1$ hyperplanes. An $\ell$-layer ReLU network partitions $\mathbb{R}^d$ into at most $\mathcal{O}(n^{(\ell-1)d})$ cells, where $n$ is the width of the network. We can enumerate these cells in $\mathcal{O}(n^{(\ell-1)d} \cdot \mathrm{poly}(N))$ time, where $N$ is the encoding size of the ReLU network (the first hidden layer gives a hyperplane arrangement; with every subsequent hidden layer, a cell is intersected with at most $n$ hyperplanes, partitioning the cell into at most $n^d$ cells).

### A.2   Additional Figures

Figure 5 illustrates how the sum of the spike function $s_{r,l}$ and penalty functions $p_r, p_l$ in the proof of Proposition 3.1 behaves for a fixed color pair.

## B   Proofs

### B.1   Proof of Theorem 2.1

*Proof.* **Zonotope (Non-)Containment.**   It is well-known that a zonotope $Z \subset \mathbb{R}^d$ with $n$ generators has $\mathcal{O}(n^{d-1})$ vertices which can be enumerated in $\mathcal{O}(n^{d-1})$ time (Ferrez et al., 2005). Note that $Z$ is contained in another zonotope $Z'$ if and only if all vertices of $Z$ are contained in $Z'$. Hence, by enumerating the vertices of $Z$ and checking containment in $Z'$ (e.g. by solving a linear program), we obtain an $\mathcal{O}(n^{d-1} \cdot \mathrm{poly}(N))$-time algorithm.

$L_p$**-Max on Zonotopes.**   We can enumerate in $\mathcal{O}(n^{d-1} \cdot \mathrm{poly}(N))$ time all vertices of the zonotope and thus obtain an $\mathcal{O}(n^{d-1} \cdot \mathrm{poly}(N) \cdot T)$ time algorithm, where $T$ denotes the time to evaluate the $L_p$-norm of a vector in $\mathbb{R}^d$.

$\square$

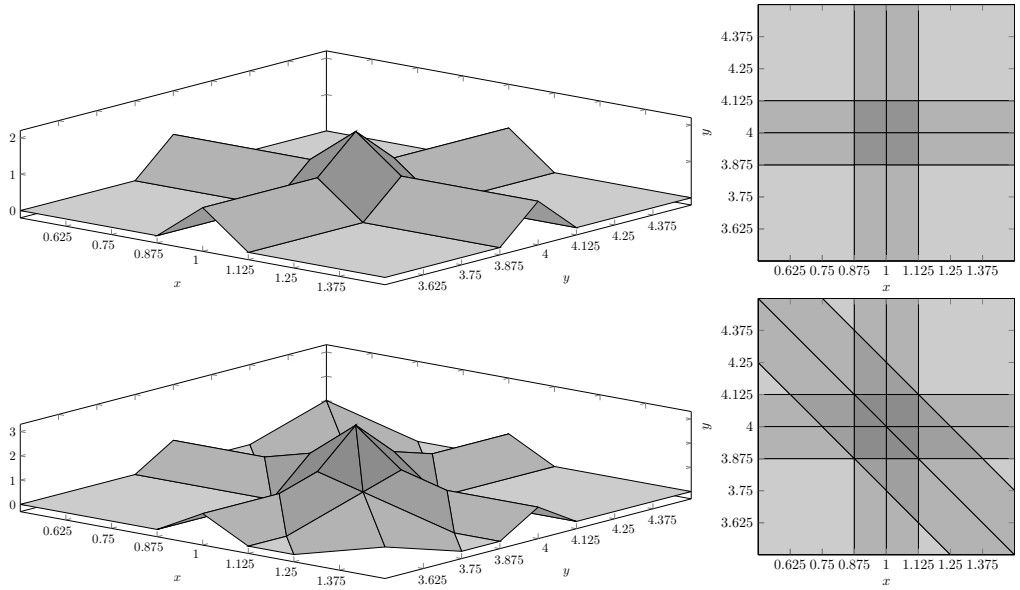

Figure 5: Illustration of the area around node values for a fixed color pair for non-adjacent and adjacent nodes. The values are based on the example given in Figure 1. For non-adjacent nodes, only the two penalty functions intersect, while for adjacent nodes, the penalty functions intersect also with the spike function.

### B.2 PROOF OF THEOREM 2.2

Before going into the proof of Theorem 2.2, we first prove that network verification for $\ell$-layer ReLU networks over polyhedra is solvable in $\mathcal{O}(n^{(\ell-1)d} \cdot \mathrm{poly}(N))$ time.

**Lemma B.1.** *Let $f\colon \mathbb{R}^d \to \mathbb{R}^m$ be an $\ell$-layer ReLU network and let $P \subseteq \mathbb{R}^d$ and $Q \subseteq \mathbb{R}^m$ be polyhedra given in halfspace representation. Then, we can decide whether $f(P) \subseteq Q$ holds in $\mathcal{O}(n^{(\ell-1)d} \cdot \mathrm{poly}(N))$ time (where $n$ is the network width and $N$ the combined encoding size of the network and the polyhedra).*

*Proof.* Let $Q = \{x \in \mathbb{R}^m : v_i^\top x \le u_i, i \in [k]\}$. Then, we have $f(P) \subseteq Q$ if and only if for every cell $C \in \Sigma_f$ and every constraint $i \in [k]$, we have

$$u_i \ge \max_{x \in C \cap P} v_i^\top f_C(x) = \max_{x \in C \cap P} v_i^\top (A_C^\top x + b_C) = v_i^\top b_C + \max_{x \in C \cap P} (A_C^\top v_i)^\top x,$$

where $f_C$ is the affine linear function of $f$ restricted to $C$, that is, $f(x) = f_C(x) := A_C^\top x + b_C$ for all $x \in C$. Note that the above condition can be verified by solving a linear program whose encoding size is polynomially bounded in $N$. Since linear programs can be solved in polynomial time and cells can be enumerated in $\mathcal{O}(n^{(\ell-1)d} \cdot \mathrm{poly}(N))$ time, it follows that we can check whether $f(P) \subseteq Q$ holds in $\mathcal{O}(n^{(\ell-1)d} \cdot \mathrm{poly}(N))$ time. $\qquad\square$

*Proof of Theorem 2.2.* Let $f\colon \mathbb{R}^d \to \mathbb{R}$ with $f(x) = W_\ell \cdot (\phi_{W_{\ell-1},b_{\ell-1}} \circ \cdots \circ \phi_{W_1,b_1})(x) + b_\ell$ be the function computed by an $\ell$-layer ReLU network and let $\Sigma_f$ be the corresponding polyhedral complex. Further, let $n$ denote the width and $N$ the encoding size of the network and let $f'\colon \mathbb{R}^d \to \mathbb{R}$ with $f'(x) := W_\ell \cdot (\phi_{W_{\ell-1},\mathbf{0}} \circ \cdots \circ \phi_{W_1,\mathbf{0}})(x)$ be the function of the ReLU network without biases.

**Computing the Maximum over a Polyhedron $P$.** Here, we assume that the maximum $\max_{x \in P} f(x)$ exists and $N$ denotes the combined encoding size of the network and the polyhedron $P$. We can compute this maximum by enumerating cells of $\Sigma_f$ and solving the linear program $\max_{x \in C \cap P} f_C(x)$ for each cell $C \in \Sigma_f$, where $f_C$ is the affine linear function of $f$ restricted to $C$, that is, $f(x) = f_C(x) := a_C^\top x + b_C$ for all $x \in C$. Note that the encoding size of the linear program is polynomially bounded in $N$. Since linear programs can be solved in polynomial time and cells can be

enumerated in $\mathcal{O}(n^{(\ell-1)d} \cdot \text{poly}(N))$ time, it follows that $\max_{x \in P} f(x) = \max\limits_{C \text{ cell of } \Sigma_f} \max\limits_{x \in C \cap P} f_C(x)$ can be computed in $\mathcal{O}(n^{(\ell-1)d} \cdot \text{poly}(N))$ time.

$\ell$**-LAYER RELU POSITIVITY.** Follows from applying Lemma B.1 to $P = \mathbb{R}^d$ and $Q = (-\infty, 0]$, since there being a point $x \in \mathbb{R}^d$ with $f(x) > 0$ is equivalent to $f(\mathbb{R}^d) \not\subseteq (-\infty, 0]$.

$\ell$**-LAYER RELU SURJECTIVITY.** Froese et al. (2025b, Lemma 14) show that for surjectivity, $f$ is surjective if and only if $f'$ is surjective, which is equivalent to there being two points $r^+, r^- \in \mathbb{R}^d$ with $f'(r^-) < 0 < f'(r^+)$. We can check this in $\mathcal{O}(n^{(\ell-1)d} \cdot \text{poly}(N))$ time by applying the algorithm for $\ell$-LAYER RELU POSITIVITY to $f'$ and $-f'$.

$\ell$**-LAYER RELU $L_p$-LIPSCHITZ CONSTANT.** Note that the $L_p$-Lipschitz constant of $f$ is equal to the maximum $L_q$-norm value (where $L_q$ is the dual norm of the $L_p$-norm, so $1/p + 1/q = 1$) of any gradient of the linear function that arises by restricting $f$ to a cell of $\Sigma_f$. Thus, by enumerating all cells of $\Sigma_f$, we obtain an $\mathcal{O}(n^{(\ell-1)d} \cdot \text{poly}(N) \cdot T)$ time algorithm, where $T$ denotes the time to evaluate the $L_q$-norm of a vector in $\mathbb{R}^d$.

$\ell$**-LAYER RELU ZERO FUNCTION CHECK.** Follows from applying Lemma B.1 to $P = \mathbb{R}^d$ and $Q = \{0\}$. $\qquad\square$

### B.3 PROOF OF PROPOSITION 3.1

Before going into detail, we first introduce a useful definition and prove a preliminary result.

**Definition B.2.** A *Sidon set* is a set of positive integers $A = \{a_1, \ldots, a_m\}$ where the sums $a_i + a_j$ with $i \leq j$ are all different.

For a survey on Sidon sets, we refer to (O'Bryant, 2004). The greedy Sidon set, introduced by Mian & Chowla (1944), is recursively constructed as follows: take $a_1 = 1$, and for $n > 1$, let $a_n$ be the smallest nonnegative integer such that $\{a_1, \ldots, a_n\}$ is a Sidon set (see A005282). Stöhr (1955) noted that $a_n \in \mathcal{O}(n^3)$ holds. We note that the greedy Sidon set of size $n$ can be computed in $n^{\mathcal{O}(1)}$ time. We use the following result.

**Lemma B.3.** *Let $A$ be a Sidon set of size $n$, and let $W_1, \ldots, W_k$ be a partition of $A$ into disjoint subsets. Then, for every pair $i, j \in \binom{[k]}{2}$, the sums $a + b$ with $a \in W_i, b \in W_j$ are all different.*

*Proof.* Suppose that there are two pairs $(a, b) \neq (c, d) \in W_i \times W_j$ with $a + b = c + d$. Then, there exist elements $a_i, a_j, a_r, a_l \in A$ such that $a_i = a, a_j = b, a_r = c, a_l = d$ with $\{i, j\} \neq \{r, l\}$ and $a_i + a_j = a_r + a_l$, which contradicts the fact that $A$ is a Sidon set. $\qquad\square$

In other words, given an element $w \in W_i + W_j$, there is exactly one pair $(w_i, w_j) \in W_i \times W_j$ with $w = w_i + w_j$.

*Proof of Proposition 3.1.* Let $(G = (V = V_1 \dot\cup \ldots \dot\cup V_k), E), k)$ be an instance of MULTICOLORED CLIQUE, where $V_c = \{v_{c,1}, \ldots, v_{c,n_c}\}$ for $c \in [k]$ and $E = \bigcup_{(r,l) \in \binom{[k]}{2}} E_{r,l}$, where $E_{r,l}$ denotes the set of edges whose nodes have color $r$ and $l$. Further, let $A$ be the greedy Sidon set of size $|V|$ and let $W_1, \ldots, W_k$ be a partition of $A$ into $k$ disjoint subsets such that $|W_i| = n_i$ holds. Note that this allows us to assign each node $v_{c,i}$ to a unique element $\omega_{c,i}$ of $A$, namely the $i$-th element of $W_c$. For every edge $\{v_{r,i}, v_{l,j}\}$, we define the constant $\omega_{r,i,l,j} := \omega_{r,i} + \omega_{l,i}$. Since $A$ is a Sidon set, the value $\omega_{r,i,l,j}$ uniquely determines the edge $\{v_{r,i}, v_{l,j}\}$. We construct a ReLU network with $k$ input variables $x_1, \ldots, x_k$ and $3(|V| + |E|)$ hidden neurons as follows.

For every color $c \in [k]$, we introduce a *node selection* gadget which ensures that the input value $x_c$ encodes a node in $V_c$. To this end, we create a "penalty function" $p_c \colon \mathbb{R} \to [0, 1]$ (see Figure 2) that has $n_c$ narrow spikes around the value $\omega_{c,i}$ (that is, it goes up from 0 to 1 and down to 0 again) for each $v_{c,i} \in V_c$ and is zero everywhere else:

$$p_c(x) := \begin{cases} 8(x - \omega_{c,i} + \frac{1}{8}), & \text{if } x \in [\omega_{c,i} - \frac{1}{8}, \omega_{c,i}], i \in [n_c] \\ 1 - 8(x - \omega_{c,i}), & \text{if } x \in (\omega_{c,i}, \omega_{c,i} + \frac{1}{8}], i \in [n_c] \\ 0, & \text{if } x \notin \bigcup_{i \in V_c} [\omega_{c,i} - \frac{1}{8}, \omega_{c,i} + \frac{1}{8}] \end{cases}.$$

The penalty function $p_c$ can be implemented with $3n_c$ hidden neurons:

$$p_c(x) = \sum_{i \in [n_c]} \left( \max\left(0, 8\left(x - \omega_{c,i} + \frac{1}{8}\right)\right) - \max(0, 16(x - \omega_{c,i})) + \max\left(0, 8\left(x - \omega_{c,i} - \frac{1}{8}\right)\right) \right).$$

Next, we introduce an *edge verification* gadget which verifies that each pair of nodes selected by the node selection gadgets is connected by an edge. For every pair of colors $(r,l) \in \binom{[k]}{2}$, we define a "spike function" $s_{r,l} \colon \mathbb{R}^2 \to [0,1]$ (see Figure 1) that is zero everywhere except for a set of $|E_{r,l}|$ parallel stripes in which $s_{r,l}$ forms a spike.

$$s_{r,l}(x,y) := \begin{cases} 4(x + y - \omega_{r,i,l,j} - \frac{1}{4}), & \text{if } x + y \in [\omega_{r,i,l,j} - \frac{1}{4}, \omega_{r,i,l,j}], \{v_{r,i}, v_{l,j}\} \in E_{r,l} \\ 1 - 4(x + y - \omega_{r,i,l,j}), & \text{if } x + y \in (\omega_{r,i,l,j}, \omega_{r,i,l,j} + \frac{1}{4}], \{v_{r,i}, v_{l,j}\} \in E_{r,l} \\ 0, & \text{if } x + y \notin \bigcup_{\{v_{r,i}, v_{l,j}\} \in E_{r,l}} [\omega_{r,i,l,j} - \frac{1}{4}, \omega_{r,i,l,j} + \frac{1}{4}] \end{cases}.$$

Note that Lemma B.3 implies that $s_{r,l}(x,y) \leq 1$ holds for any input $(x,y) \in \mathbb{R}^2$, as the sums $\omega_{r,i} + \omega_{l,i} = \omega_{r,i,l,j}$ for $\{v_{r,i}, v_{l,j}\} \in E_{r,l}$ are all integral and different. Thus, the spike functions attain value 1 if and only if its inputs correspond to two nodes that share an edge. The spike function can be implemented with $3|E_{r,l}|$ hidden neurons:

$$s_{r,l}(x,y) = \sum_{\{v_{r,i}, v_{l,j}\} \in E_{r,l}} \left( \max\left(0, 4\left(x + y - \omega_{r,i,l,j} + \frac{1}{4}\right)\right) - \max(0, 8(x + y - \omega_{r,i,l,j})) \right.$$

$$\left. + \max\left(0, 4\left(x + y - \omega_{r,i,l,j} - \frac{1}{4}\right)\right) \right).$$

By computing all penalty and spike functions in parallel and summing them up at the output neuron, we obtain a ReLU network that computes the function $f \colon \mathbb{R}^k \to \mathbb{R}$ with

$$f(x_1, \ldots, x_k) = \sum_{(r,l) \in \binom{[k]}{2}} s_{r,l}(x_r, x_l) + \sum_{c \in [k]} p_c(x_c).$$

Since every spike and penalty function is lower bounded by 0 and upper bounded by 1, it follows that $f$ is lower bounded by 0 and upper bounded by $k + \binom{k}{2}$.

First, we show that the existence of a $k$-colored clique in $G$ implies $\max_{x \in \mathbb{R}^k} f(x) = k + \binom{k}{2}$. Suppose that $\{v_{1,a_1}, \ldots, v_{k,a_k}\} \subset V$ forms a $k$-colored clique in $G$. Then, we claim that the point $x^* = (\omega_{1,a_1}, \ldots, \omega_{k,a_k})$ is a point with $f(x^*) = k + \binom{k}{2}$. First, note that $p_c(x_c^*) = 1$ holds for all $c \in [k]$. Further, for each pair of colors $(r,l) \in \binom{[k]}{2}$, $s_{r,l}(x_r^*, x_l^*) = 1$ holds, since $\{v_{r,a_r}, v_{l,a_l}\}$ is an edge in $E_{rl}$. Thus, we have $f(x^*) = k + \binom{k}{2}$.

Now, we show that if there is a point $x^* \in \mathbb{R}^k$ with $f(x^*) > k + \binom{k}{2} - 1$, then $G$ has a $k$-colored clique. Suppose that $x^* \in \mathbb{R}^k$ is such a point. For this to be the case, the output of all spike and penalty functions must be strictly greater than zero, that is, we have $p_c(x_c^*) > 0$ for every $c \in [k]$ and $s_{r,l}(x_r^*, x_l^*) > 0$ for every pair $(r,l) \in \binom{[k]}{2}$, since otherwise $f(x^*) \leq k + \binom{k}{2} - 1$ holds. For every $c \in [k]$, $p_c(x_c^*) > 0$ implies by definition that there is exactly one element $a_c \in V_c$ with $x_c^* \in (\omega_{c,a_c} - \frac{1}{8}, \omega_{c,a_c} + \frac{1}{8})$. In other words, the input $x_c^*$ corresponds to the node $v_{c,a_c}$. We now claim that $\{v_{1,a_1}, \ldots, v_{k,a_k}\}$ forms a $k$-colored clique in $G$. To see this, observe that for every pair $(r,l) \in \binom{[k]}{2}$, $s_{r,l}(x_r^*, x_l^*) > 0$ together with $x_r^* + x_l^* \in (\omega_{r,a_r,l,a_l} - \frac{1}{4}, \omega_{r,a_r,l,a_l} + \frac{1}{4})$ implies by definition of $s_{r,l}$ that $\{v_{r,a_r}, v_{l,a_l}\}$ is an edge, which proves that $\{v_{1,a_1}, \ldots, v_{k,a_k}\}$ is indeed a $k$-colored clique. $\square$

### B.4 PROOF OF THEOREM 4.2

Before proving Theorem 4.2, we first prove an auxiliary lemma.

**Lemma B.4.** *Let* $f\colon \mathbb{R}^d \to \mathbb{R}$, $\quad f(x) = \sum_{i=1}^n c_i \max\{0, a_i^\top x + b_i\} + B$ *be the function that is computed by a 2-layer ReLU network, where* $a_i, b_i, c_i, B$ *are the weights and biases of this network, and let* $h\colon \mathbb{R}^{d+1} \to \mathbb{R}$ *be the function computed by the homogenization of this network. Then, we have* $h(x, 1) = h(-x, -1)$ *if and only if* $\sum_{i=1}^n c_i(a_i^\top x + b_i) = 0$.

*Proof.* We have $h(x, y) = \sum_{i=1}^n c_i \max\{0, a_i^\top x + b_i y\} + B|y|$ and thus

$$h(x,1) - h(-x,-1) = \sum_{i=1}^n c_i(\max\{0, a_i^\top x + b_i\} - \max\{0, -(a_i^\top x + b_i)\}) = \sum_{i=1}^n c_i(a_i^\top x + b_i).$$

$\square$

*Proof of Theorem 4.2.* We give a parameterized reduction from MULTICOLORED CLIQUE. Let $(G = (V = V_1 \dot\cup \ldots \dot\cup V_k, E), k)$ be an instance of MULTICOLORED CLIQUE, and let $f\colon \mathbb{R}^k \to \mathbb{R}$ be the function of the network constructed in the proof of Proposition 3.1. Next, we modify the network by setting the bias of the output node to $1 - k - \binom{k}{2}$. Let $g\colon \mathbb{R}^k \to \mathbb{R}$ be the function of this modified network and let $h\colon \mathbb{R}^{k+1} \to \mathbb{R}$ be the function computed by the homogenization of this modified network. By construction, we have $h(x, 1) = g(x) = f(x) + 1 - k - \binom{k}{2}$ for every $x \in \mathbb{R}^k$. Note that $h(-x, -1) = h(x, 1)$ holds for every $x \in \mathbb{R}^k$, which follows directly from the definition of $f$ and Lemma B.4. Since the ReLU network corresponding to $h$ has no biases, $h$ is positively homogeneous and thus $h(\lambda x, \lambda y) = \lambda h(x, y)$ holds for every $\lambda \geq 0$.

By Proposition 3.1, $G$ has a $k$-colored clique if and only if $g$ has a *positive point*, since $\max_{x \in \mathbb{R}^k} g(x) = 1$ holds if $G$ has a $k$-colored clique and $\max_{x \in \mathbb{R}^k} g(x) \leq 0$ otherwise. To finish the proof, observe that $h$ has a positive point if and only if $g$ has a positive point. If $g$ has a positive point $x^*$, then $h$ also has a positive point $(x^*, 1)$. On the other hand, if $h$ has a positive point $(x^+, y^+)$, then by positive homogeneity $\mathrm{sgn}(y^+) \cdot \frac{x^+}{|y^+|}$ is a positive point for $g$, since $0 < \frac{1}{|y^+|} h(x^+, y^+) = h(\frac{x^+}{|y^+|}, \mathrm{sgn}(y^+)) = g(\mathrm{sgn}(y^+) \cdot \frac{x^+}{|y^+|})$. Note that $y^+ = 0$ is not possible, since $h(x, 0) = 0$ for every $x \in \mathbb{R}^k$ by construction, as deleting biases leads to the cancellation of all terms in the spike and penalty functions. $\square$

## B.5 PROOF OF COROLLARY 4.3

*Proof.* We give a parameterized reduction from MULTICOLORED CLIQUE to approximating the maximum of a 2-layer ReLU network. Let $(G = (V = V_1 \dot\cup \ldots \dot\cup V_k, E), k)$ be an instance of MULTICOLORED CLIQUE, let $f\colon \mathbb{R}^k \to \mathbb{R}$ be the function of the network constructed in the proof of Proposition 3.1 and let $g\colon \mathbb{R}^k \to \mathbb{R}$ be the function of the same network with an additional bias of $1 - k - \binom{k}{2}$ at the output node, that is, $g(x) = f(x) + 1 - k - \binom{k}{2}$ holds for every $x \in \mathbb{R}^k$. With Proposition 3.1, it follows that we have $\max_{x \in \mathbb{R}^k} g(x) = 1$ if and only if $G$ has a $k$-colored clique and $\max_{x \in \mathbb{R}^k} g(x) \leq 0$ otherwise. Thus, approximating the maximum of this network within any multiplicative factor over the polytope $P = [0, n^3]^d$ would allow us to distinguish between the two cases, which implies the theorem. $\square$

## B.6 PROOF OF THEOREM 5.1

*Proof.* Follows from Theorem 4.2 and the equivalence to 2-LAYER RELU POSITIVITY without biases (Froese et al., 2025b, Proposition 18). $\square$

## B.7 PROOF OF THEOREM 6.2

*Proof.* We give a parameterized reduction (which is also a polynomial reduction) from MULTICOLORED CLIQUE to 2-LAYER RELU $L_p$-LIPSCHITZ CONSTANT. We first discuss the case $p \in [1, \infty]$ and later discuss which modifications are necessary to extend the proof to $p \in (0, 1)$.

Let $(G = (V = V_1 \dot\cup \ldots \dot\cup V_k, E), k)$ be an instance of MULTICOLORED CLIQUE. Further, let $g\colon \mathbb{R}^{k+1} \to \mathbb{R}$ be the function computed by the homogenized network constructed in the proof of Proposition 3.1.

Note that for any positively homogeneous CPWL function $f : \mathbb{R}^d \to \mathbb{R}$, the $L_p$-Lipschitz constant can be rewritten to $L_p(f) = \max_{\|x\|_p \leq 1} |f(x)|$, which follows from the fact that $L_p(f)$ is the maximum $L_p$-Lipschitz constant of $f$ restricted to any of the full-dimensional cones $C \in \Sigma_f$, where $f(x) = a_C^\top x$ for all $x \in C$ ($f$ is linear on $C$) and the $L_p$-Lipschitz constant of the linear function in this cell is equal to $\max_{\|x\|_p \leq 1} |a_c^\top x|$ (Jordan & Dimakis, 2020).

We now scale all $y$ coefficients of $g$ by $\varepsilon := \frac{1}{2k \cdot a_n \cdot (k + \binom{k}{2})}$, where $a_n \in \mathcal{O}(n^3)$ is the maximum element of the greedy Sidon set of size $n$, and obtain the modified positively homogeneous CPWL function $h : \mathbb{R}^{k+1} \to \mathbb{R}$. Now, every maximizer $x^*$ of $\max_{x \in \mathbb{R}^k} h(x, 1)$ satisfies $|x_i^*| \leq a_n \cdot \varepsilon$, since every maximizer $x'$ of $\max_{x \in \mathbb{R}^k} g(x, 1)$ previously satisfied $|x_i'| \leq a_n$. This follows from the fact that scaling the $y$ coefficients is equivalent to scaling the spike and penalty functions in the reduction. Now, we define

$$\mathcal{L} := \max_{x \in \mathbb{R}^k} h(x, 1)$$

and claim the following:

$$\mathcal{L} \geq L_p(h) \geq \mathcal{L}(1 - k \cdot a_n \cdot \varepsilon).$$

The inequality $\mathcal{L} \geq L_p(h)$ follows from the fact that if $(x^*, y^*) \in \arg\max_{\|(x,y)\|_p \leq 1} h(x, y)$, then $|y^*| \leq 1$ and

$$L_p(h) = h(x^*, y^*) = |y^*| \cdot h\left(\frac{x^*}{|y^*|}, \operatorname{sgn}(y^*)\right) \leq h\left(\frac{x^*}{|y^*|}, \operatorname{sgn}(y^*)\right) \leq \max_{x \in \mathbb{R}^k} h(x, 1) = \mathcal{L}$$

holds. The second inequality follows from the fact that if $x^*$ is a maximizer of $\max_{x \in \mathbb{R}^k} h(x, 1)$, then $|x_i^*| \leq a_n \cdot \varepsilon$ and thus

$$\|(1 - k \cdot a_n \cdot \varepsilon) \cdot (x^*, 1)\|_p \leq \|(1 - k \cdot a_n \cdot \varepsilon) \cdot (x^*, 1)\|_1 \leq (1 - k \cdot a_n \cdot \varepsilon) + \sum_{i=1}^k |x_i^*| \leq 1$$

holds, which makes $(1 - k \cdot a_n \cdot \varepsilon) \cdot (x^*, 1)$ a feasible point for $\max_{\|(x,y)\|_p \leq 1} h(x, y)$.

Given this estimation, we now make a case distinction: if $G$ has a $k$-colored clique, then $\mathcal{L} = (k + \binom{k}{2}) \cdot \varepsilon$ and

$$L_p(h) \geq (1 - k \cdot a_n \cdot \varepsilon) \cdot \left(k + \binom{k}{2}\right) \cdot \varepsilon = \left(1 - \frac{1}{2(k + \binom{k}{2})}\right) \cdot \left(k + \binom{k}{2}\right) \cdot \varepsilon = \left(k + \binom{k}{2} - \frac{1}{2}\right) \cdot \varepsilon.$$

On the other hand, if $G$ does not have a $k$-colored clique, then

$$L_p(h) \leq \mathcal{L} \leq \left(k + \binom{k}{2} - 1\right) \cdot \varepsilon.$$

Therefore, we have a separation of the $L_p$-Lipschitz constant $L_p(h)$ depending on whether $G$ has a $k$-colored clique or not. With this, the 2-LAYER RELU $L_p$-LIPSCHITZ CONSTANT instance consisting of $L = (k + \binom{k}{2} - \frac{1}{2}) \cdot \varepsilon$ and the underlying network of $h$ is a yes-instance if and only if $G$ is a yes-instance of MULTICOLORED CLIQUE, which concludes the proof.

For every $p \in (0, 1) \cap \mathbb{Q}$, we can scale the network with $\varepsilon := \frac{1}{a_n \cdot k^N} \cdot \left(p\left(1 - \frac{k + \binom{k}{2} - \frac{1}{2}}{k + \binom{k}{2}}\right)\right)^N$, where $N = \lceil 1/p \rceil$. Note that since $p$ is a fixed rational constant, $\varepsilon$ is also rational and still polynomial in the input size. We estimate

$$\varepsilon = \frac{1}{a_n \cdot k^N} \cdot \left(p\left(1 - \frac{k + \binom{k}{2} - \frac{1}{2}}{k + \binom{k}{2}}\right)\right)^N \leq \frac{1}{a_n \cdot k^N} \cdot \left(1 - \left(\frac{k + \binom{k}{2} - \frac{1}{2}}{k + \binom{k}{2}}\right)^p\right)^N$$

$$\leq \frac{1}{a_n \cdot k^{1/p}} \cdot \left(1 - \left(\frac{k + \binom{k}{2} - \frac{1}{2}}{k + \binom{k}{2}}\right)^p\right)^{1/p} =: \varepsilon^*.$$

Next, we can estimate $\mathcal{L} \geq L_p(h) \geq \mathcal{L}(1 - k \cdot (a_n \cdot \varepsilon)^p)^{1/p}$, where the second inequality follows from the fact that if $x^*$ is a maximizer of $\max_{x \in \mathbb{R}^k} h(x, 1)$, then $|x_i^*| \leq a_n \cdot \varepsilon$ and thus

$$\|(1 - k \cdot (a_n \cdot \varepsilon)^p)^{1/p} \cdot (x^*, 1)\|_p = \left(1 - k \cdot (a_n \cdot \varepsilon)^p + (1 - k \cdot (a_n \cdot \varepsilon)^p) \sum_{i=1}^k |x_i^*|^p\right)^{1/p}$$

$$\leq (1 - k \cdot (a_n \cdot \varepsilon)^p + (1 - k \cdot (a_n \cdot \varepsilon)^p) \cdot k \cdot a_n \cdot \varepsilon)^{1/p} \leq 1$$

holds, which makes $(1 - k \cdot (a_n \cdot \varepsilon)^p)^{1/p} \cdot (x^*, 1)$ a feasible point for $\max_{\|(x,y)\|_p \leq 1} h(x, y)$.

We then proceed with the estimation for the case where $G$ has a $k$-colored clique with

$$L_p(h) \geq (1 - k \cdot (a_n \cdot \varepsilon)^p)^{1/p} \cdot \left(k + \binom{k}{2}\right) \cdot \varepsilon$$

$$\geq (1 - k \cdot (a_n \cdot \varepsilon^*)^p)^{1/p} \cdot \left(k + \binom{k}{2}\right) \cdot \varepsilon$$

$$= \frac{k + \binom{k}{2} - \frac{1}{2}}{k + \binom{k}{2}} \cdot \left(k + \binom{k}{2}\right) \cdot \varepsilon = \left(k + \binom{k}{2} - \frac{1}{2}\right) \cdot \varepsilon,$$

which gives the same estimation as previously for $p \in [1, \infty]$ (note that we cannot directly use $\varepsilon^*$ as scaling factor, since $\varepsilon^*$ might not be rational). □

## B.8 PROOF OF THEOREM 7.1

*Proof.* Let $A = (a_1, \ldots, a_n) \in \mathbb{R}^{d \times n}$ be a matrix and let $Z(A) = \sum_{i=1}^n \mathrm{conv}\{0, a_i\} \subset \mathbb{R}^d$ be the corresponding zonotope. Defining $c := \frac{1}{2} \sum_{i=1}^n a_i$ as the center of the zonotope, we have

$$Z - c = \sum_{i=1}^n \mathrm{conv}\left\{-\frac{a_i}{2}, \frac{a_i}{2}\right\} = \sum_{i=1}^n \mathrm{conv}\left\{0, -\frac{a_i}{2}\right\} + \sum_{i=1}^n \mathrm{conv}\left\{0, \frac{a_i}{2}\right\} \subset \mathbb{R}^d.$$

We now construct the matrix $B = \left(\frac{a_1}{2}, \ldots, \frac{a_n}{2}\right)^\top \in \mathbb{R}^{n \times d}$. Then, we have that

$$\|Bx\|_1 = \sum_{i=1}^n \left|\frac{a_i}{2}^\top x\right| = \sum_{i=1}^n \max\left\{0, -\frac{a_i}{2}^\top x\right\} + \sum_{i=1}^n \max\left\{0, \frac{a_i}{2}^\top x\right\}$$

is the support function of the zonotope $Z - c$.

Applying the polynomial algorithm of Cohen & Peng (2015), we find a matrix $B' = (b_1', \ldots, b_r')^\top \in \mathbb{R}^{r \times d}$ with $r \in \mathcal{O}(d \log d \varepsilon^{-2})$ such that with high probability, $(1 + \varepsilon)^{-1}\|Bx\|_1 \leq \|B'x\|_1 \leq (1 + \varepsilon)\|Bx\|_1$ holds for all $x \in \mathbb{R}^d$. From the duality between zonotopes and their support function, this implies $(1 + \varepsilon)^{-1} Z((B^\top, -B^\top)) \subseteq Z - c \subseteq (1 + \varepsilon) Z((B^\top, -B^\top))$. Defining $A' = (2b_1' + c, \ldots, 2b_r' + c) \in \mathbb{R}^{d \times r}$, it follows that $(1 + \varepsilon)^{-1}\|Bx\|_1 \leq \|B'x\|_1 \leq (1 + \varepsilon)\|Bx\|_1$ implies $(1 + \varepsilon)^{-1} Z(A') \subseteq Z(A) \subseteq (1 + \varepsilon) Z(A')$, which implies the theorem. □