# OpenReview forum: "Parameterized Hardness of Zonotope Containment and Neural Network Verification"
_ICLR.cc/2026/Conference — ICLR 2026 Poster_

### Official Review · Reviewer_Lmnz · 2025-10-20

**Soundness:** 3
**Presentation:** 3
**Contribution:** 2
**Rating:** 6
**Confidence:** 4

**Summary:**

This paper studies the parameterized complexity of zonotope containment and some problems related to neural network verification. It extends the prior work and proves the strongest complexity bounds so far. The results shows that under Exponential Time Hypothesis (ETH), brute-force enumeration is asymptotically optimal in complexity.

**Strengths:**

The studies problem is relevant, i.e., providing better complexity characterization for certain neural network certification related problems.

The paper is clearly and rogirously written.

The results are novel and extend the scientific boundary.

**Weaknesses:**

It is not clear how these results might influence practice. While knowing the complexities might be interesting, it seems to have minimal impact and guidance on neural network certification practice.

Some related work in the difficulty of neural network is missing: [1] shows that for a two-dimensional simple convex monotone CPWL function, there exists no single-neuron relaxations that can precisely certify any network encoding the function; [2] extends the result and shows that every convex relaxation is incomplete in neural network certification. These works study the theoretical boundary of neural network certification in the practical algorithms, while this work characterizes the difficulty of the general problem.

[1] https://arxiv.org/abs/2311.04015

[2] https://arxiv.org/abs/2410.06816

**Questions:**

Please address the weaknesses.

---

> ### Author Response · Authors · 2025-11-20
>
> Thank you for your careful reading and evaluation of our paper, and for your constructive feedback.
> We believe we have addressed all raised questions in the responses below, and we are happy to answer any further questions.
>
> >**Weaknesses:**
> >
> >It is not clear how these results might influence practice. While knowing the complexities might be interesting, it seems to have minimal impact and guidance on neural network certification practice.
>
> We discuss practical implications of our results in our general response above.
>
> >Some related work in the difficulty of neural network is missing: [1] shows that for a two-dimensional simple convex monotone CPWL function, there exists no single-neuron relaxations that can precisely certify any network encoding the function; [2] extends the result and shows that every convex relaxation is incomplete in neural network certification. These works study the theoretical boundary of neural network certification in the practical algorithms, while this work characterizes the difficulty of the general problem.
> >
> >[1] https://arxiv.org/abs/2311.04015
> >
> >[2] https://arxiv.org/abs/2410.06816
>
> Thank you for bringing these references to our attention. We now discuss and cite them in our related work.

---

> > ### Comment · Reviewer_Lmnz · 2025-11-23
> >
> > Thanks for the new information. All my concerns are now clear.

---

### Official Review · Reviewer_Druf · 2025-10-23

**Soundness:** 3
**Presentation:** 3
**Contribution:** 3
**Rating:** 6
**Confidence:** 3

**Summary:**

This paper presents new theoretical results on the complexity of the problem of proving properties of ReLU networks. Specifically, the authors study the parameterized complexity of checking for positivity, checking for surjectivity, and computing the Lipschitz constant of ReLU networks and show that these problems are W[1]-hard. The analysis of these properties centers around mapping 2-layer ReLU networks to the multicolored clique problem. The authors also show how these results can be extrapolated to the zonotope containment problem, which has implications in other domains as well.

**Strengths:**

- The theoretical complexity results developed in the paper are the strongest known so far and resolve an open problem.
- The authors develop results for multiple different neural network verification tasks.
- The results rely on and have implications for well-studied problems in other domains (multicolored clique and zonotope containment).

**Weaknesses:**

It is not clear from the paper the practical implications of the derived theory and complexity results. Therefore, it may not be immediately useful for practitioners working on neural network verification tools. Many of the high-performance neural network verification tools used these days rely on overapproximations, and it is unclear how these results might inform decisions for further development in this domain.

**Questions:**

- What are the practical implications of the complexity results?
- What should we as a community take from these results as we continue to progress the field of neural network verification?
- Do these results have any implications for approximate neural network verification techniques?

---

> ### Author Response · Authors · 2025-11-20
>
> Thank you for your careful reading and evaluation of our paper, and for your constructive feedback.
> We believe we have addressed all raised questions in the responses below, and we are happy to answer any further questions.
>
> >**Weaknesses:**
> >
> >It is not clear from the paper the practical implications of the derived theory and complexity results. Therefore, it may not be immediately useful for practitioners working on neural network verification tools. Many of the high-performance neural network verification tools used these days rely on overapproximations, and it is unclear how these results might inform decisions for further development in this domain.
>
> We discuss practical implications of our results in our general response above, as well as implications for approximate techniques in our response to your dedicated question below.
>
> >**Questions:**
> >
> >What are the practical implications of the complexity results?
>
> We discuss practical implications of our complexity results in our general response above.
>
> >What should we as a community take from these results as we continue to progress the field of neural network verification?
>
> We discuss some takeaways in our general response above.
>
> >Do these results have any implications for approximate neural network verification techniques?
>
> We give a partial answer to this in our general comment above.
> Our results also imply some approximation hardness results, such as for approximating the maximum of a ReLU network on a bounded domain.
> At the same time, our results show that a compromise is unavoidable: to obtain reasonable running times, one must rely on heuristics and approximate techniques, since exact verification is (in general) not achievable within a reasonable runtime.

---

> > ### Comment · Reviewer_Druf · 2025-11-25
> >
> > I acknowledge the rebuttal and thank the authors for replying to my questions!

---

### Official Review · Reviewer_4wHP · 2025-10-29

**Soundness:** 3
**Presentation:** 3
**Contribution:** 3
**Rating:** 6
**Confidence:** 4

**Summary:**

The paper presents a collection of (parameterised) complexity results related
to feedforward neural network (FNN) verification tasks, as well as zonotope
containment.

The paper presents the following core results:
- 2-Layer ReLU Positivity: The problem of deciding whether a given 2-layer FNN
  with ReLU activations can produce an output greater than 0 is W[1]-hard.
- 2-Layer ReLU Surjectivity: The problem of deciding whether a given 2-layer FNN
  with ReLU activations computes a surjective function is W[1]-hard.
- Zonotope Non-Containment: The problem of deciding whether, for two given
  zonotopes, one is not included in the other is W[1]-hard.
- 2-Layer ReLU $L_p$-Lipschitz Constant: The problem of deciding whether the
  $L_p$ Lipschitz constant of a given FNN with ReLU activations is greater than
  some given threshold is NP- and W[1]-hard.

Here, the class W[1] is the first level of the W-hierarchy widely considered in
the theory of parameterised complexity.

**Strengths:**

The paper presents its results in a rigorous and convincing manner,
providing compelling insights into the complexity of FNN verification-related
problems.

Furthermore, the structure and craftsmanship of the paper are commendable and feel very polished.

**Weaknesses:**

The paper could improve in several areas:
- While reading, it sometimes feels like a collection of varied results, somewhat loosely
  related to FNN verification. Providing more explanation on why these problems are of
  interest would be beneficial.
- In the same light, the preliminaries seem to lack a proper definition of the class W[1].
  It is assumed that many readers may not be fully acquainted with parameterised complexity,
  which might lead to some confusion.

**Questions:**

- Can you provide more details regarding the implications for actual
          verification tasks, such as positivity, given your W[1]-hardness result?
- Could you elaborate on why problems such as positivity, surjectivity,
          or the Lipschitz constant are of significance to the verification
          community? They seem somewhat abstract or overly simplistic.

---

> ### Author Response · Authors · 2025-11-20
>
> Thank you for your careful reading and evaluation of our paper, and for your constructive feedback.
> We believe we have addressed all raised questions in the responses below, and we are happy to answer any further questions.
>
> >**Weaknesses:**
> >
> >The paper could improve in several areas:
> >- While reading, it sometimes feels like a collection of varied results, somewhat loosely related to FNN verification. Providing more explanation on why these problems are of interest would be beneficial.
>
> For general comments on relevance of our results, please refer to our general response above.
> For explanations on specific problems, we address these in our response to the dedicated question below.
>
> >- In the same light, the preliminaries seem to lack a proper definition of the class W[1]. It is assumed that many readers may not be fully acquainted with parameterised complexity, which might lead to some confusion.
>
> The class W[1] is originally defined to contain all parameterized problems for which there is a parameterized reduction to Weighted $1$-Normalized SAT (see Downey and Fellows 2013).
> Alternatively, one could simply define it as all problems reducible to Clique parameterized by the clique size. We have added this definition to our preliminaries.
>
> >**Questions:**
> >
> >Can you provide more details regarding the implications for actual verification tasks, such as positivity, given your W[1]-hardness result?
>
> We discuss practical implications of our results in our general response above.
> For a more intuitive explanation of what our hardness result for positivity implies for the more general verification problem, see our response to the following question.
>
> > Could you elaborate on why problems such as positivity, surjectivity, or the Lipschitz constant are of significance to the verification community? They seem somewhat abstract or overly simplistic.
>
> Positivity is one of the simplest cases of network verification (and in fact equivalent to surjectivity). Since even this basic problem is hard to decide, it follows that the general verification problem cannot be easier, since it includes the basic problem of positivity. In other words, hardness for a simple problem is a stronger statement than hardness for a more general problem. In addition, the hardness of positivity implies hardness of verification for every input set $X$ that contains a ball, as shown in https://arxiv.org/abs/2405.19805. Thus, the computational hardness persists independently of any assumptions on the input set $X$.
> Moreover, positivity can be phrased as a Zonotope Containment problem (or norm maximization over zonotopes), which is a fundamental problem in computational geometry with a wide range of applications.
>
> The Lipschitz-constant is a widely studied aspect of neural networks and can also be phrased in terms of maximizing a norm over a zonotope.
> Knowing the Lipschitz-constant can be useful for verification: by evaluating a ReLU network $f:\mathbb{R}^d\to \mathbb{R}^m$ on a point $x'$, the Lipschitz-constant $L_p(f)$ restricts which outputs can be achieved for other inputs $x$, since $\|\|f(x)-f(x')\|\|_p\leq L_p(f)\cdot  \|\|x-x'\|\|_p$ holds for all $x\in \mathbb{R}^d$.
> This information might be useful for specific verification tasks.

---

> > ### Comment · Reviewer_4wHP · 2025-11-25
> >
> > Thank you for the clarifications. Already in my initial review, I considered the paper a meaningful contribution to our understanding of the complexity aspects of exact neural-network verification.
> >
> > Since my concerns regarding the definition of W[1] and the broader impact have now been addressed in the revised version, I am satisfied with the changes. I therefore support acceptance and have adjusted my score accordingly.

---

### Official Review · Reviewer_wktp · 2025-11-01

**Soundness:** 4
**Presentation:** 4
**Contribution:** 3
**Rating:** 8
**Confidence:** 4

**Summary:**

The paper uses the tools of parameterized complexity to prove stronger hardness results for several problems arising within the study of ReLU neural networks (for instance, computing the $L_p$-Lipschitz constant of the network), or that can be linked to it (such as Zonotope Containment).

The paper relies on parameterized reductions, and more precisely on one main reduction using the W[1]-hard problem Multicolored Clique as a starting point and constructing a 2-layer ReLU network such that the maximum value of the function computed by the network determines whether the Multicolored Clique instance had a solution or not.

After obtaining the W[1]-hardness and sometimes ETH-conditional lower bounds for some problems, the authors also use known equivalences or reductions between problems to spread these results and get a more complete picture. As they remark in the conclusion, the common denominator to these problems is that they correspond in a way to maximizing a norm over a zonotope.

**Strengths:**

The theoretical analysis is sound, and yields new results for a variety of problems.

Gives a theoretical insight into why the naive enumeration algorithms are somehow optimal for these problems, and fosters more application of parameterized complexity to the field.

For the $L_p$-Lipschitz constant, the hardness result for the exact computation of 2-layer networks is complemented by a hardness result for the approximation of the  $L_p$-Lipschitz constant of 3-layer networks.

The paper is well written.

**Weaknesses:**

It is not clear whether some properties of real-world networks make it possible to circumvent these worst-case theorems. However, I agree this is unavoidable for a theoretic analysis, and the identification of such empirical properties is not the topic of the paper.

**Questions:**

l141: this means that for $\\ell=1$, no ReLU neuron is involved?

l269: why do you need two extra hidden neurons for the bias of the output?

l426: what is the size of the LPs?



Suggestions and Typos:

l21, abstract: For readability, consider "Moreover, we show that (a): ....., (b): ....., and (c): ...... are all NP-hard and W[1]-hard ....".

l174: $L\\in \\mathbb{R}$, right?

Figure 1: the use of "r, l" as colors is technically fine, but it is on a psychological level confusing since it looks like "right, left", with the "right" being always on the left, in the figure and in expressions. Not necessary strictly speaking, but it might be nice swapping them.

l404: please reformulate, highlighting that the maximum is over linear regions.

---

> ### Author Response · Authors · 2025-11-20
>
> Thank you for your careful reading and evaluation of our paper, and for your constructive feedback.
> We believe we have addressed all raised questions in the responses below, and we are happy to answer any further questions.
>
> >**Weaknesses:**
> >
> >It is not clear whether some properties of real-world networks make it possible to circumvent these worst-case theorems. However, I agree this is unavoidable for a theoretic analysis, and the identification of such empirical properties is not the topic of the paper.
>
> We discuss this point in our general response above.
> We believe that it is an interesting direction for further research to explore if the hardness results also hold for weights that are learned by training algorithms such as gradient descent.
>
> >**Questions:**
> >
> >l141: this means that for $\ell=1$, no ReLU neuron is involved?
>
> We use standard terminology and define the number of layers $\ell$ as the number of hidden layers plus one (the input layer is not counted).
> Thus, a 1-layer ReLU network ($\ell=1$) has no hidden layer - and no ReLU neurons - and computes only an affine linear function.
> Most of our hardness results are proved for 2-layer networks, i.e., networks with a single hidden layer.
> In particular, this means that these results are depth-optimal, since the corresponding problems become easy when no ReLU neurons are involved.
>
> >l269: why do you need two extra hidden neurons for the bias of the output?
>
> We set the output bias of the homogenized network to zero and instead use two extra hidden neurons to propagate the term $|y|\cdot b = b \cdot\max(0,y)+b\cdot\max(0, -y)$ to the output node (where $b$ is the output bias of the original network), since this term is required in our construction.
>
> >l426: what is the size of the LPs?
>
> As stated in the proof, the encoding size of the polytope $Q$ (given in halfspace representation, i.e, through linear inequalities) and the projection $\pi$ is polynomial in the encoding size of the ICNN network $N$.
> Since all LPs are of the form $\max_{y\in Q}c^\top \pi(y)$, the size of each LP is itself polynomial in the encoding size of the network $N$.
> To be more precise, by the results of Hertrich \& Loho 2024, the number of constraints in each LP is equal to the number of neurons in the network times two.
>
> >Suggestions and Typos:
> >
> >l21, abstract: For readability, consider "Moreover, we show that (a): ....., (b): ....., and (c): ...... are all NP-hard and W[1]-hard ....".
>
> We have incorporated this suggestion into the new version.
>
> >l174: $L\in \mathbb{R}$, right?
>
> Yes, we have added $L\in \mathbb{R}$ to the sentence before the problem descriptions in the new version.
>
> >Figure 1: the use of "r, l" as colors is technically fine, but it is on a psychological level confusing since it looks like "right, left", with the "right" being always on the left, in the figure and in expressions. Not necessary strictly speaking, but it might be nice swapping them.
>
> To avoid confusion, we have changed Figure 1 in the new version.
>
> >l404: please reformulate, highlighting that the maximum is over linear regions.
>
> We have reformulated the text to highlight this in the new version.

---

> > ### Comment · Reviewer_wktp · 2025-11-27
> >
> > I thank the authors for the clarifications. I will continue supporting acceptance.

---

### Author Response · Authors · 2025-11-20
**General response**

With this response, we aim to comprehensively address several related questions:

>- Can some properties of real-world networks make it possible to circumvent these worst-case theorems?
>- What are the (practical) implications for actual verification tasks given these results?
>- What can practitioners learn from this?
>- Do these results have implications for approximate techniques?

Our hardness result foremost shows that there is no hope to achieve substantially faster exact verification algorithms than the naive enumeration based approach. That is, the exponent in the running time has to depend linearly on the input dimension. So there is no algorithm with a polynomial running time of fixed degree, e.g., there is no linear-time algorithm that works for every constant dimension.

Moreover, it does not help to assume that the network weights are sparse and small since our construction uses only a constant number of (polynomially bounded) non-zero weights for each ReLU neuron.

It is thus not easy to formulate a general guidance to circumvent this hardness in practice.
One would have to make very specific assumptions on the network structure which ensure that the number of linear regions is small and easy to enumerate (it is not clear which assumptions would be natural here and whether networks trained on real-world data satisfy them).
Alternatively, one might use techniques (possibly incorporated into the training process) that guarantee efficient verification or use special architectures (such as ICNNs).

To sum up, our results prove and justify that such techniques and the use of heuristics (such as over-approximations based on convex relaxations) are indeed required in practice to achieve reasonable runtimes.

We appreciate the reviewers push for clarity on these questions.
We have thus incorporated (parts of) this answer into our conclusion to more clearly highlight these aspects.

---

### Meta-Review · Area_Chair_f3Hh · 2025-12-29

**Summary:**

This paper presents new theoretical results on the complexity of several problems related to the verification of shallow ReLU networks, including checking for positivity, checking for surjectivity, and computing the Lipschitz constant. The paper shows that these problems are W[1]-hard and naive enumeration algorithms are somehow optimal. All reviewers agree that the paper is well-written, the theory is rigorous and convincing, and they provide compelling insights.

However, reviewers also raised common concerns or questions regarding the practical implications of the purely theoretical results in this paper. The authors referred the reviewers to the general response which basically says techniques used in recent verifiers (heuristics, overapproximation, etc.) are indeed necessary.

The AC feels this critically limits the significance of this work. The hardness of exact verification for ReLU networks has been well-known since a long time ago (e.g., in the 2017 paper "Reluplex: An Efficient SMT Solver for Verifying Deep Neural Networks" (not cited by the authors)). Showing an even stronger hardness result has very limited significance given that the hardness shown more than 8 years ago is already sufficiently strong to prohibit straightforward implementations. Indeed, recent NN verification works do not pursue exact verification and do not work on arbitrary ReLU networks. Models that are actually amenable to verification are produced by special training algorithms such as adversarial training, training with regularizers, or certified training that can bring certain properties to ease verification. Recent NN verification algorithms have been commonly utilizing overapproximations for many years. As such, the settings studied by this work are very outdated and do not look informative for practitioners developing recent techniques.

The AC personally feels this paper is below the acceptance bar due to limited significance, but would respect the opinion of the reviewers to recommend acceptance.

**Reviewer Concerns:**

The common major concern is that the practical implications look minimal (see above). The authors added some discussions in the general response. While they are good for clarification purposes, they do not materially address the concern.

There were some minor concerns such as clarifying preliminaries and adding related works, which have been addressed.

**Reviewer Scores:**

All reviewers gave positive scores and explicitly responded to the authors:
* Reviewer wktp: likely would maintain their score of 8 as they mentioned to continue supporting acceptance
* Reviewer 4wHP: mentioned their score adjustment which is likely from 6 to 8\.
* Reviewer Druf: would likely maintain their score of 6 as they acknowledged the rebuttal
* Reviewer Lmnz: would likely maintain their score of 6 as they acknowledged the rebuttal

---

### Decision · Program_Chairs · 2026-01-26

Accept (Poster)